# Emerging climate impact on carbon sinks in a consolidated carbon budget

Pierre Friedlingstein[1,2 ✉], Corinne Le Quéré[3], Michael O'Sullivan[1], Judith Hauck[4,5], Peter Landschützer[6], Ingrid T. Luijkx[7], Hongmei Li[8,9], Auke van der Woude[7], Clemens Schwingshackl[10], Julia Pongratz[9,10], Pierre Regnier[11], Robbie M. Andrew[12], Dorothee C. E. Bakker[13], Josep G. Canadell[14], Philippe Ciais[15], Thomas Gasser[15,16], Matthew W. Jones[3], Xin Lan[17,18], Eric Morgan[19], Are Olsen[20,21], Glen P. Peters[12], Wouter Peters[7,22], Stephen Sitch[1] & Hanqin Tian[23]

Despite the adoption of the Paris Agreement 10 years ago, carbon dioxide ($CO_2$) emissions from burning fossil fuels continue to increase, pushing atmospheric $CO_2$ levels to 423 ppm in 2024 and driving human-induced warming to 1.36 °C, within years of breaching the 1.5 °C limit[1,2]. Accurate reporting of anthropogenic and natural $CO_2$ sources and sinks is a prerequisite to tracking the effectiveness of climate policy and detecting carbon-sink responses to climate change. Yet notable mismatches between reported emissions and sinks have so far prevented confident interpretation of their trends and drivers[1]. Here we present and integrate recent advances in observations and process understanding to address some long-standing issues in global carbon budget estimates. We show that the magnitude of the natural land sink is substantially smaller than previously estimated, whereas net emissions from anthropogenic land-use change are revised upwards[1]. The ocean sink is 15% larger than the land sink, consistent with recent evidence from oceanic and atmospheric observations[3,4]. Climate change reduces the efficiency of the sinks, particularly on land, contributing 8.3 ± 1.4 ppm to the atmospheric $CO_2$ increase since 1960. The combined effects of climate change and deforestation have turned Southeast Asian and large parts of South American tropical forests from $CO_2$ sinks to sources. This underscores the need to halt deforestation and limit warming to prevent further loss of carbon stored on land. Improved confidence in assessments of $CO_2$ sources and sinks is fundamental for effective climate policy.

The increase in atmospheric carbon dioxide ($CO_2$) concentration has been systematically monitored since the late 1950s, marking the beginning of comprehensive research into the global carbon cycle[5]. It soon became evident that the observed increase in atmospheric $CO_2$ was smaller than the $CO_2$ emissions from burning fossil fuels, indicating that terrestrial ecosystems and/or the ocean acted as carbon sinks[6]. Until the late 1980s, it was believed that the ocean was the main sink of carbon, whereas the role of land ecosystems was unclear and was often referred to as the 'missing sink'[7]. The presence of a large $CO_2$ sink on land was confirmed later on, supported by field studies[8], biomass inventories[9] or vegetation modelling[10]. Over the past 20 years, our understanding of the global carbon cycle has rapidly improved, supported by the annual assessments of the global carbon budget (GCB) activity of the Global Carbon Project. This activity has enabled continuous community review of the anthropogenic perturbation of the global carbon cycle[1,11]. The GCB assessments are widely used in science and policy, including in the latest assessment of the Intergovernmental Panel on Climate Change[12].

The carbon balance among individual components of the global carbon cycle provides a rigorous test of our understanding of the

[1]Faculty of Environment, Science and Economy, University of Exeter, Exeter, UK. [2]Laboratoire de Météorologie Dynamique, Institut Pierre-Simon Laplace, CNRS, Ecole Normale Supérieure, Université PSL, Sorbonne Université, Ecole Polytechnique, Paris, France. [3]Tyndall Centre for Climate Change Research, School of Environmental Sciences, University of East Anglia, Norwich, UK. [4]Alfred-Wegener-Institut, Helmholtz-Zentrum für Polar- und Meeresforschung, Bremerhaven, Germany. [5]Faculty of Biology/Chemistry, Universität Bremen, Bremen, Germany. [6]Flanders Marine Institute (VLIZ), Ostend, Belgium. [7]Environmental Sciences Group, Dept of Meteorology and Air Quality, Wageningen University, Wageningen, The Netherlands. [8]Helmholtz-Zentrum Hereon, Geesthacht, Germany. [9]Max Planck Institute for Meteorology, Hamburg, Germany. [10]Department of Geography, Ludwig-Maximilians-Universität München, Munich, Germany. [11]Department of Geoscience, Environment and Society-BGEOSYS, Université Libre de Bruxelles, Brussels, Belgium. [12]CICERO Center for International Climate Research, Oslo, Norway. [13]Centre for Ocean and Atmospheric Sciences, School of Environmental Sciences, University of East Anglia, Norwich, UK. [14]CSIRO Environment, Canberra, Australian Capital Territory, Australia. [15]Laboratoire des Sciences du Climat et de l'Environnement, LSCE/IPSL, CEA-CNRS-UVSQ, Université Paris-Saclay, Gif-sur-Yvette, France. [16]International Institute for Applied Systems Analysis (IIASA), Laxenburg, Austria. [17]Cooperative Institute for Research in Environmental Sciences (CIRES), University of Colorado Boulder, Boulder, CO, USA. [18]National Oceanic and Atmospheric Administration Global Monitoring Laboratory (NOAA/GML), Boulder, CO, USA. [19]Scripps Institution of Oceanography, University of California San Diego, La Jolla, CA, USA. [20]Geophysical Institute, University of Bergen, Bergen, Norway. [21]Bjerknes Centre for Climate Research, Bergen, Norway. [22]University of Groningen, Centre for Isotope Research, Groningen, The Netherlands. [23]Center for Earth System Science and Global Sustainability, Schiller Institute for Integrated Science and Society, Department of Earth and Environmental Sciences, Boston College, Chestnut Hill, MA, USA. ✉e-mail: p.friedlingstein@exeter.ac.uk

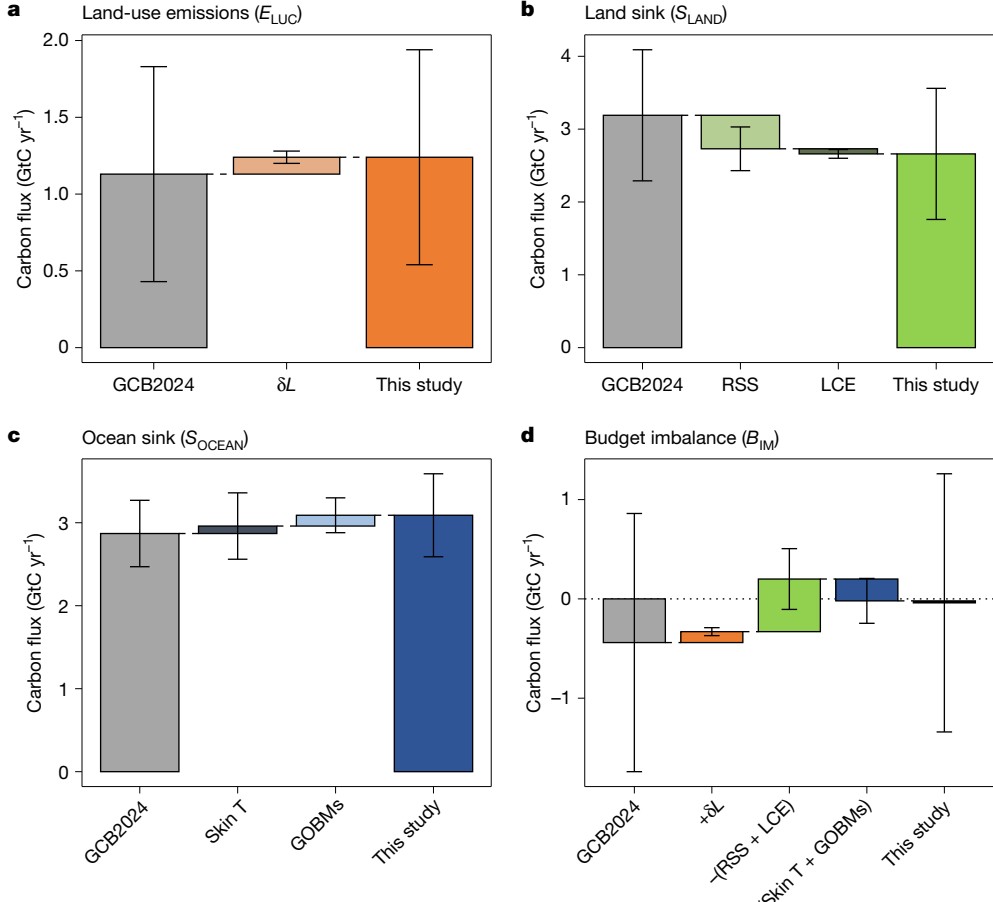

**Fig. 1 | Revised components of the GCB. a**, Net land-use emissions ($E_{LUC}$). **b**, Land sink ($S_{LAND}$). **c**, Ocean sink ($S_{OCEAN}$). **d**, Budget imbalance ($B_{IM}$). The grey bars on the left show the GCB2024 estimate, the intermediate bars show the incremental corrections from this study, and the coloured bars on the right show the consolidated estimates. Components are averaged over the past

decade (2014–2023). $\delta L$, RSS, LCE and Skin T refer to the transient carbon densities correction, the replaced sinks and sources correction, the lateral carbon export correction and the ocean cool skin temperature correction, respectively (Methods). Error bars are 1 standard deviation uncertainty.

carbon cycle: mass conservation implies that estimated net emissions from fossil ($E_{FOS}$) and land-use change ($E_{LUC}$) and uptake by the ocean and land sinks ($S_{OCEAN}$ and $S_{LAND}$) must balance the observation-based atmospheric $CO_2$ growth rate $G_{ATM}$ perfectly. This has not been the case throughout the history of the GCB reports, including in the latest 2024 update[13] (hereafter GCB2024). GCB2024 reported a budget imbalance ($B_{IM}$; $B_{IM} = E_{FOS} + E_{LUC} - S_{LAND} - S_{OCEAN} - G_{ATM}$) over the past decade of $-0.4 \pm 1.4$ GtC yr$^{-1}$, which is about 10% of the observation-based atmospheric $CO_2$ growth rate. Despite its large uncertainty, the negative $B_{IM}$ implies that estimated sources were too low and/or estimated sinks too large. Over the past 65 years, the $B_{IM}$ also showed a negative trend of $-0.14 \pm 0.04$ GtC yr$^{-1}$ per decade, statistically significant at the 1% level ($P = 0.003$), with a positive $B_{IM}$ in the early part of the record and a negative $B_{IM}$ in the most recent years (Extended Data Fig. 1).

A statistically significant trend in the $B_{IM}$ impedes robust interpretation of trends in individual components of the GCB. Hence, reducing the magnitude and trend of the $B_{IM}$ is a prerequisite to reliably assessing temporal changes in the strength of the carbon sinks. Here we present and integrate recent advances in observations and process understanding to improve our estimates of components of the GCB, with direct impact on the magnitude and trend of the $B_{IM}$. These improvements allow a more robust assessment of the human interference on the global carbon cycle over the past 65 years, and of the emerging impacts of climate change on the evolution of the carbon sinks.

## Introducing the latest evidence

The net land-use change $CO_2$ emissions ($E_{LUC}$) assessed in the GCB are derived from bookkeeping models forced by reported changes in land use. Most bookkeeping models assume that land-cover types, such as forest or pasture, have distinct but static equilibrium carbon densities (that is, amount of carbon per unit area of a full-grown ecosystem)[13]. This assumption allows to isolate the direct land-use impact (for example, owing to deforestation, afforestation) from indirect human-induced effects on vegetation[14,15] such as higher global biomass and higher soil carbon densities owing to environmental effects (for example, owing to atmospheric $CO_2$ increase)[16]. However, neglecting the effects of environmental changes in $E_{LUC}$ estimates results in an underestimation of the historical $E_{LUC}$ trend[16,17]. To address this issue, we replaced the static carbon densities used in bookkeeping models by transient values informed by dynamic global vegetation model (DGVM)-derived carbon dynamics[17,18] (Methods). Accounting for transient carbon densities leads to an increase in net $E_{LUC}$ of $0.11 \pm 0.04$ GtC yr$^{-1}$ over the past decade, and additional emissions of $3.0 \pm 1.0$ GtC since 1960 (Fig. 1a and Extended Data Fig. 2b).

The land $CO_2$ sink ($S_{LAND}$) is estimated in the GCB from DGVMs using historical simulations that assume a constant pre-industrial land cover. In doing so, the models do not double account for $CO_2$ fluxes associated with land-cover changes from anthropogenic land use, which are already included in $E_{LUC}$. However, given the historical reduction in

**Table 1 | Global carbon budget as in GCB2024 and consolidated budget from this study**

| | $G_{ATM}$ | $E_{FOS}$ | $E_{LUC}$ | $S_{LAND}$ | Net land | $S_{OCEAN}$ | $B_{IM}$ |
|---|---|---|---|---|---|---|---|
| GCB2024 | 5.2±0.02 | 9.7±0.5 | 1.1±0.7 | 3.2±0.9 | 2.1±1.1 | 2.9±0.4 | −0.4±1.3 |
| This study | 5.2±0.02 | 9.7±0.5 | 1.2±0.7 | 2.7±0.9 | 1.4±1.1 | 3.1±0.5 | −0.02±1.3 |
| Difference | 0 | 0 | +0.1 | −0.5 | −0.6 | +0.2 | +0.4 |
| Atmospheric inversions | 5.2±0.0 | 9.7±0.5 | NA | NA | 1.4±0.5 | 3.1±0.5 | 0 |
| Atmospheric O₂ | 5.2±0.0 | 9.7±0.5 | NA | NA | 1.0±0.8 | 3.4±0.5 | 0 |

'Net land' is the net land $CO_2$ flux, calculated as $S_{LAND}-E_{LUC}$. Atmospheric inversions and atmospheric $O_2$ do provide 'Net land' but do not separate $E_{LUC}$ from $S_{LAND}$. The budget imbalance ($B_{IM}$) is the difference between anthropogenic net emissions ($E_{FOS}+E_{LUC}$) and accumulation of carbon in the atmosphere, land and ocean ($G_{ATM}+S_{LAND}+S_{OCEAN}$). By design, atmospheric inversions and atmospheric $O_2$ budget imbalance is null. The uncertainty represents ±1 s.d. as in ref. 1. Annual $CO_2$ fluxes are averaged over the 2014–2023 decade. Units are GtC yr⁻¹. NA, not available.

forest cover and expansion of agriculture, assuming a pre-industrial land cover leads to an overestimation of the land sink[17–20]. This is a known bias now referred to as the replaced sinks and sources (RSS)[17,19,21]. To address this issue, we developed a new correction method using outputs from the DGVMs that resolve net land–atmosphere carbon fluxes at the plant-functional-type level (Methods). Accounting for evolving land-cover change leads to a decrease of the mean $S_{LAND}$ by $0.5 \pm 0.3$ GtC yr⁻¹ over the past decade, and a decrease of 21 GtC since 1960 (Fig. 1b and Extended Data Fig. 3d).

The land and ocean $CO_2$ sinks in the GCB account for the lateral carbon export (LCE) from land ecosystems to inland waters, coastal environments and the open ocean using natural (pre-industrial) estimates of $0.65 \pm 0.30$ GtC yr⁻¹ (refs. 22,23) but neglecting its anthropogenic perturbation. Recent advances in understanding aquatic carbon cycle processes indicate an increase in carbon exported from terrestrial ecosystems to the aquatic environment, with an increased outgassing of $CO_2$ from these aquatic systems to the atmosphere, increased carbon storage in aquatic sediments and export to the ocean[24,25] (Methods). Accounting for the anthropogenic perturbation of LCE leads to a decrease of the mean $S_{LAND}$ by $0.07 \pm 0.06$ GtC yr⁻¹ over the past decade (Fig. 1b and Extended Data Fig. 3).

The ocean $CO_2$ sink in the GCB combines independent estimates from data products based on observations ($fCO_2$ products, where $fCO_2$ is the fugacity of $CO_2$)[26,27] as well as global ocean biogeochemical models (GOBMs). $fCO_2$ products and GOBMs broadly agree on ocean sink trends and variability, with remaining differences mostly explained by limited data and seasonal biased sampling causing overestimation in the decadal trends of $fCO_2$ products, and possible GOBM underestimation of decadal variability[28], especially in the Southern Ocean[29–31]. However, $fCO_2$ products suggest a substantially larger ocean sink than GOBMs ($3.1 \pm 0.3$ GtC yr⁻¹ versus $2.6 \pm 0.4$ GtC yr⁻¹, respectively, over 2014–2023), which is also supported by independent constraints derived from atmospheric $CO_2$ and oxygen ($O_2$) observations[3] as well as ocean interior observations[4]. Multiple model evaluation efforts have now shown that GOBMs underestimate the mean oceanic sink on the order of 10%, based on evidence of too weak overturning circulation[32], ocean interior constraints[33] and biases arising from spin-up strategies[34]. In parallel, estimates from $fCO_2$ products could also be biased low because they do not account for temperature gradients between the measurement depth, usually several metres below the surface, and the surface skin layer where the gas exchange takes place[35–37]. Accounting for the GOBMs bias and for skin temperatures and the warm layer in $fCO_2$ products leads to an increased $S_{OCEAN}$ of $0.2 \pm 0.23$ GtC yr⁻¹ over the past decade, and an increase of $11 \pm 14$ GtC since 1960 (Fig. 1c and Extended Data Fig. 2c).

$CO_2$ emissions from fossil fuels ($E_{FOS}$) include the oxidation of fossil fuels from combustion, chemical reactions, decomposition of fossil carbonates and the $CO_2$ uptake from the cement carbonation[1]. The GCB estimate of $E_{FOS}$ ($9.7 \pm 0.5$ GtC yr⁻¹ for the 2014–2023 period) is a composite of different datasets, aimed to give the best emission estimate and reduce biases. The differences between independent datasets are well understood, with the range between different datasets around 5% and with all showing similar trends[38]. $E_{FOS}$ misses minor emission sources in some developing countries for decomposition of some carbonates, estimated to be <0.5% of the global total. The cement carbonation sink is probably the most poorly constrained element of $E_{FOS}$, but at 0.2 GtC yr⁻¹ in recent years, the contribution to $E_{FOS}$ uncertainty is small. Hence, we do not have any compelling reason to suspect a substantial bias in the global $E_{FOS}$ mean or trend that would require a correction in this study.

The atmospheric $CO_2$ growth rate ($G_{ATM}$) in the GCB is based on marine-boundary-layer $CO_2$ mole fraction observations (in ppm yr⁻¹), which have only a small measurement uncertainty[39]. These measurements are subsequently converted to mass growth rates in GtC yr⁻¹ using a conversion factor, which so far has been assumed to be a constant value of 2.124 GtC ppm⁻¹, without associated uncertainty[40]. However, the surface fluxes that lead to changes in atmospheric mole fractions are not instantaneously observed at the surface stations, given that atmospheric mixing takes time. The surface network is also not fully representative of the whole atmosphere[41]. Any variability and uncertainty in the conversion factor would propagate into the estimated annual $CO_2$ growth rate ($G_{ATM}$) and its uncertainty. Here we quantify the annual conversion-factor values and their uncertainties using the atmospheric inversions from the GCB (Methods). In Extended Data Fig. 4, we show these conversion factors and the resulting uncertainty on $G_{ATM}$ and the $B_{IM}$. Including annually varying conversion factors would mainly reduce the variability of the $B_{IM}$ (up to 40%) but has no effect on its mean or trend. This interannual effect of the conversion factor will be further evaluated and considered for inclusion in future GCB assessments.

## Consolidating the GCB

The inclusion of known missing processes and the associated corrections on $E_{LUC}$, $S_{LAND}$ and $S_{OCEAN}$ in the GCB2024 estimate[1] results in a consolidated GCB (Table 1, and Extended Data Tables 1 and 2). The revised estimate of $E_{LUC}$, when accounting for transient carbon densities, is $1.2 \pm 0.7$ GtC yr⁻¹ for the past decade (2014–2023). Although the correction increases land-use-change emissions with time, the statistically significant decline in $E_{LUC}$ of 0.2 GtC per decade since the late 1990s, as identified in GCB2024, remains ($P < 0.001$). About 75% of the $0.11 \pm 0.04$ GtC yr⁻¹ increase in $E_{LUC}$ is due to larger net land-use-change emissions in South America, Southeast Asia and Africa. It is noted that although the net effect of anthropogenic land-use change is a source of $CO_2$ to the atmosphere, parts of the world, including North America, Europe and China, are currently net carbon sinks from land-use change. Total global anthropogenic net $CO_2$ emissions ($E_{FOS} + E_{LUC}$) increased until the 2000s but remained relatively constant after 2010 at around 11 GtC yr⁻¹.

$S_{LAND}$ is substantially reduced when accounting for evolving land-cover change and for the increase in terrestrial carbon outgassed by inland waters. The revised mean land sink is $2.7 \pm 0.9$ GtC yr⁻¹ over 2014–2023 (Fig. 1b and Table 1). As a result, the revised net land $CO_2$ flux ($S_{LAND} - E_{LUC}$) is reduced by 31% from a sink of $2.1 \pm 1.1$ GtC yr⁻¹ to a

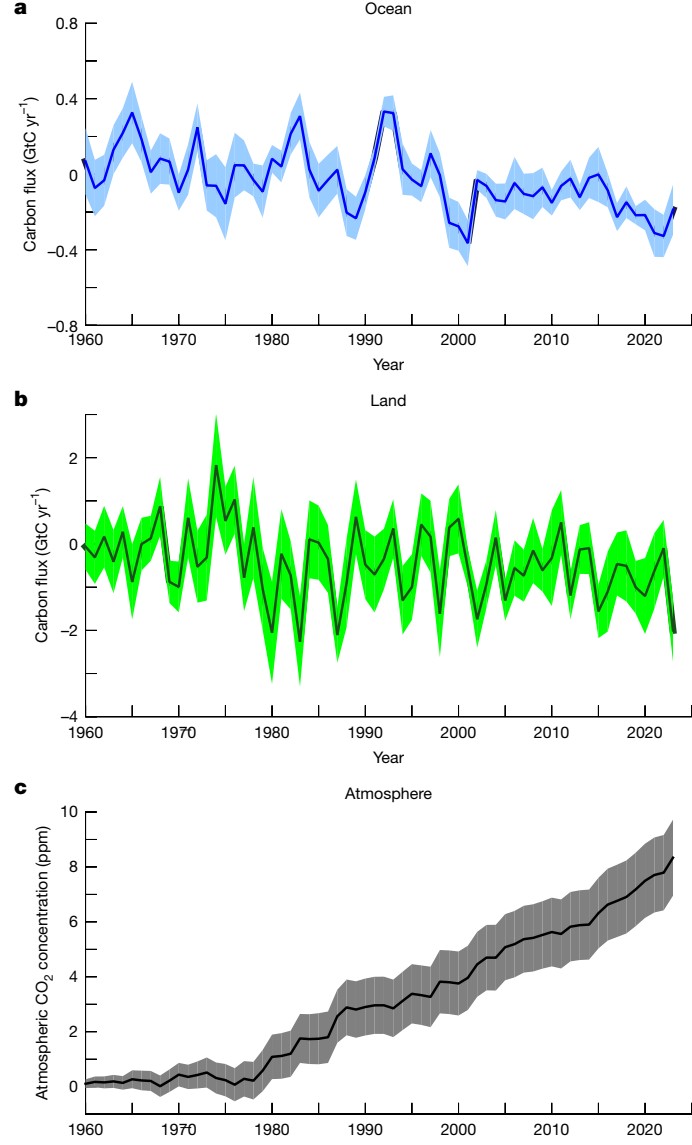

**Fig. 2 | Impact of climate change on carbon sinks and atmospheric CO₂ increase. a–c**, Impact of climate change on the ocean sink ($S_{OCEAN}$) as simulated by GOBMs (**a**), the land sink ($S_{LAND}$) as simulated by DGVMs (**b**), and their cumulative effect on the atmospheric CO₂ concentration increase since 1960 (**c**).

sink of $1.4 \pm 1.1$ GtC yr$^{-1}$ (Table 1). Conversely, the revised ocean CO₂ sink is increased by 8% when accounting for the effect of the warm layer and cool skin on ocean $f$CO₂ products and correcting for the known GOBMs bias, reaching $3.1 \pm 0.5$ GtC yr$^{-1}$ over the past decade (Fig. 1c and Table 1). As a result of these revisions, the ocean sink is about 15% larger than the land sink whereas it was 10% lower in GCB2024 (Table 1), although these differences remain within the uncertainty bounds of both fluxes.

The corrections applied to $E_{LUC}$, $S_{LAND}$ and $S_{OCEAN}$ are each within the uncertainty of the initial estimates; hence, the revised estimates are not statistically significantly different from the GCB2024 estimates (Table 1). However, the corrections applied here are based on known biogeochemical processes, which have not been considered in the GCB estimates so far. Furthermore, high confidence can be placed on the sign of each of these corrections: assuming constant vegetation densities leads to an underestimation of $E_{LUC}$, assuming pre-industrial land cover leads to an overestimation of $S_{LAND}$, ignoring historical increase in lateral carbon export also leads to an overestimation of $S_{LAND}$, and neglecting the ocean cool-skin effect leads to an underestimation of $S_{OCEAN}$. Hence

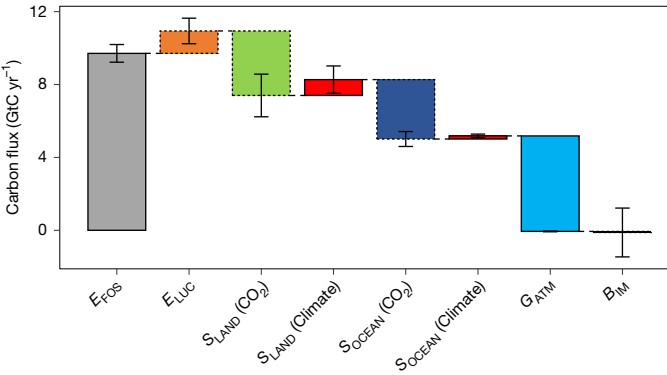

**Fig. 3 | Consolidated GCB.** CO₂ emissions from fossil fuels ($E_{FOS}$), the revised net land-use-change emissions ($E_{LUC}$), the revised land sink and ocean sink ($S_{LAND}$ and $S_{OCEAN}$) both separated into their response to CO₂ and response to climate, the atmospheric CO₂ growth rate ($G_{ATM}$), and the residual budget imbalance ($B_{IM}$). Components are averaged over the past decade (2014–2023). The dashed outlines indicate an update in this study compared with GCB2024.

the revised estimate of $E_{LUC}$, $S_{LAND}$ and $S_{OCEAN}$ represents an improvement in their representation in the GCB. Furthermore, the revised budget, with a smaller net land CO₂ ($1.4 \pm 1.2$ GtC yr$^{-1}$) and a larger ocean sink ($3.1 \pm 0.5$ GtC yr$^{-1}$), is fully consistent with the estimates from atmospheric inversions ($1.4 \pm 0.5$ GtC yr$^{-1}$ and $3.1 \pm 0.5$ GtC yr$^{-1}$ for the net land flux and the ocean sink, respectively), and with estimates derived from atmospheric O₂ observations ($1.0 \pm 0.8$ GtC yr$^{-1}$ and $3.4 \pm 0.5$ GtC yr$^{-1}$, respectively)[1,3,42] (Table 1). The convergence of these independent estimates gives stronger confidence that this revised budget provides more robust estimates compared with GCB2024.

The budget imbalance, which was $-0.4 \pm 1.3$ GtC yr$^{-1}$ over 2014–2023 in GCB2024, is reduced to near zero ($-0.1 \pm 1.3$ GtC yr$^{-1}$) (Fig. 1d and Table 1), although it is not statistically significantly different from the GCB2024 estimate. Finally, the statistically significant negative trend in the $B_{IM}$ over the past 65 years of $-0.14 \pm 0.04$ GtC per decade ($P = 0.003$) in the GCB2024 estimate is now reduced to a non-significant trend of $-0.06 \pm 0.04$ GtC per decade ($P = 0.14$), adding confidence in the revised estimate of the GCB presented here (Extended Data Fig. 2f).

## Influence of climate change

With virtually no imbalance, the consolidated GCB provides a basis for analysing the long-term evolution of the land and ocean sinks and their role in mitigating the atmospheric CO₂ increase owing to anthropogenic CO₂ emissions. Climate change is widely expected to cause a reduction of CO₂-induced land and ocean carbon sinks (relative to a theoretical case with the same atmospheric CO₂ increase but no climate change)[12,43,44]. Using additional historical simulations of GOBMs and DGVMs driven by the observed atmospheric CO₂ increase but under a constant climate forcing (Methods), we estimate that the effect of climate change has reduced the land and ocean sinks by $0.8 \pm 0.9$ GtC yr$^{-1}$ ($-23\%$) and $0.18 \pm 0.1$ GtC yr$^{-1}$ ($-6\%$), respectively over the past decade (Figs. 2a,b and 3), with tropical regions accounting for the largest effect on land (Fig. 4). The cumulative reduction in the land and ocean sinks combined amounts to $30 \pm 6$ GtC ($29 \pm 6$ GtC and $2 \pm 1$ GtC, respectively) since 1960, implying that the carbon–climate feedback has already contributed $8.3 \pm 1.4$ ppm (8%) to the increase in atmospheric CO₂ concentration (Fig. 2c).

The net land CO₂ flux can be decomposed in three contributions: the response to atmospheric CO₂ increase, the response to climate change (for example, temperature, rainfall) and land-use change (Extended Data Fig. 5). Over the decade of 2014–2023, the atmospheric CO₂ increase induced a $3.6 \pm 1$ GtC yr$^{-1}$ sink, whereas the effect of climate and

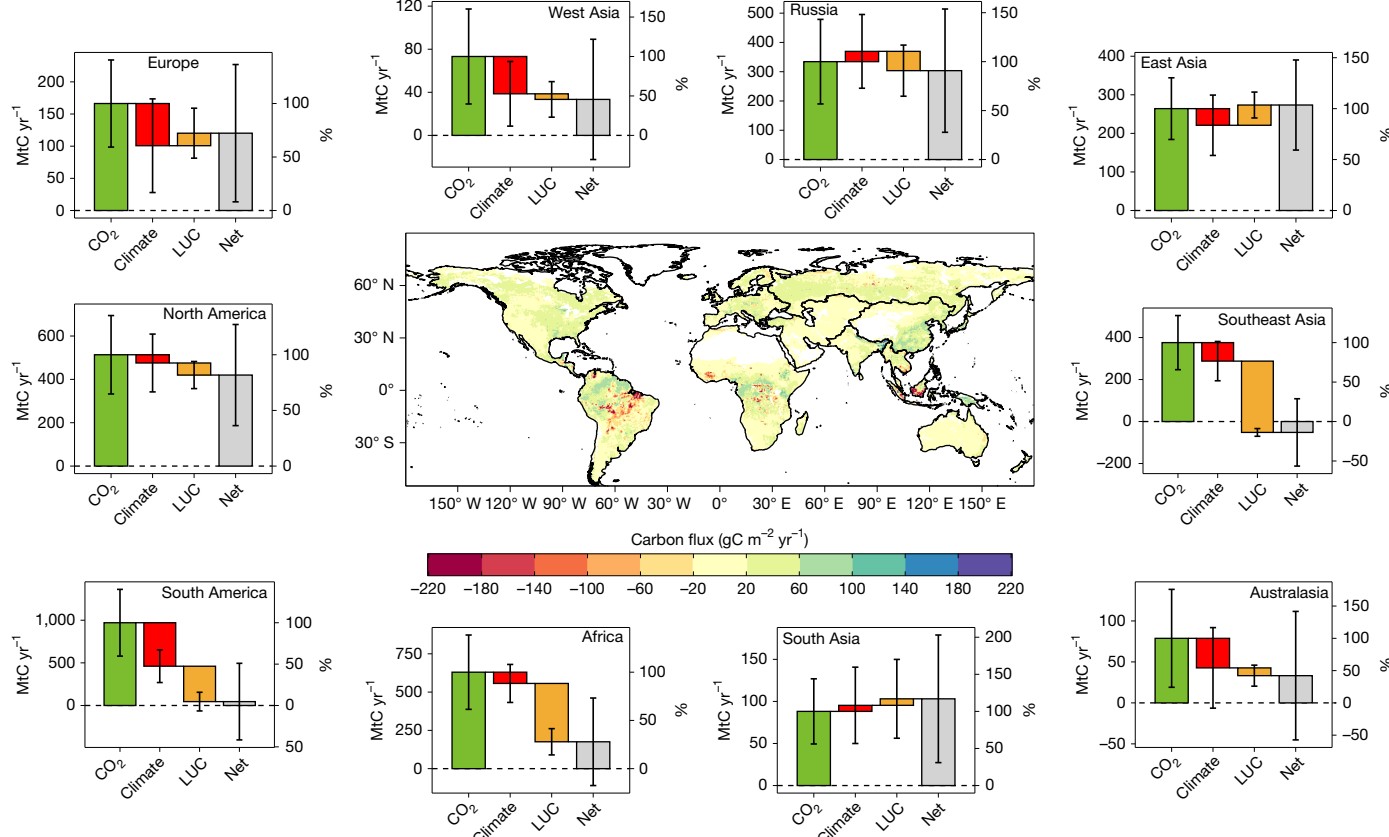

**Fig. 4 | Land CO₂ fluxes and attribution effects.** Decadal mean (2014–2023) of the net land $CO_2$ flux ($S_{LAND} - E_{LUC}$; central map and grey bars for each land RECCAP region) and attribution to the effects of atmospheric $CO_2$ increase ($CO_2$ fertilization; green bars), climate impact (red bars) and land-use change (LUC; orange bars). $CO_2$ and climate flux uncertainties are calculated as the 1σ spread among DGVMs from GCB2024. $E_{LUC}$ uncertainty is calculated as the 1σ spread among bookkeeping models from GCB2024. The uncertainty on the net flux is the square root of the sum of squares of the three component fluxes. Percentage changes (%, right axis) are relative to the $CO_2$ fertilization case (green bars).

land-use change led to a source of $0.9 \pm 0.6$ GtC yr$^{-1}$ and $1.2 \pm 0.7$ GtC yr$^{-1}$, respectively, bringing the net land $CO_2$ flux to a sink of $1.4 \pm 1.2$ GtC yr$^{-1}$. The combined effect of climate change and land-use change is largest in the tropics. Although deforestation is the main driver of carbon losses in Africa and Southeast Asia, climate impacts on ecosystems are the dominant causes of carbon losses in South America (Fig. 4), in line with observational evidence[45,46]. Our findings reinforce the need to halt deforestation and to mitigate climate change to prevent an increasingly larger fraction of the terrestrial biosphere from becoming a source of $CO_2$.

## Implications

Recent advances in observations and understanding implemented here within the GCB have contributed to addressing some of the long-standing issues and improving coherence between bottom-up estimates from DGVMs and GOBMs and top-down estimates based on atmospheric $CO_2$ inversions and $O_2$ observations. Important uncertainties remain, as reflected by the large interannual variability still present in the $B_{IM}$, and global agreement between bottom-up and top-down estimates could still be owing to compensating errors in critical processes in components of the GCB. Further improvements are required in several areas, including on the estimates of carbon losses from land degradation; the understanding of the long-term impact of fires on carbon storage; the representation of small-scale physical processes in GOBMs; the understanding of the variability of the biological ocean carbon pump; the Southern Ocean observational coverage for better $f$CO₂-product representation; and the

reconciliation of bottom-up and top-down estimates at the regional level. Delivering on these issues hinges on continued monitoring of atmospheric and surface-ocean $CO_2$ levels, which are fundamental to carbon cycle research. Maintaining regular assessments of the sources and sinks of $CO_2$ and integrating the latest understanding will facilitate monitoring changes in the natural carbon cycle and lead to more informed and effective decisions.

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

## Methods

### Land-use change emissions and transient carbon densities correction

In the GCB, $E_{LUC}$ is estimated based on four bookkeeping models driven by historical land-use-change data. All but one of the bookkeeping models (OSCAR, see below) use static equilibrium carbon density values for vegetation and soil from various sources, representative of 'present day' carbon densities. The OSCAR bookkeeping model does not require any adjustment as it already endogenously simulates changes in biome carbon densities under environmental changes, in parallel to the bookkeeping calculation of $E_{LUC}$ (refs. 18,47). Although not used in GCB2024, the BLUE bookkeeping model also offers alternative $E_{LUC}$ estimates based on transient carbon densities[17]. To adjust for $\delta L$ (the transient carbon densities) in BLUE, the static equilibrium carbon densities are converted into transient densities based on the carbon density evolution from DGVMs from the GCB (under simulations with transient environmental changes but constant land cover, termed S2; see below). Transient biomass carbon densities are derived based on 12 DGVMs and transient soil carbon densities based on 7 DGVMs providing the necessary providing the necessary plant-functional-type (PFT)-level output.

For the other two bookkeeping models that use static carbon densities in GCB2024 (H&C23 and LUCE), the $E_{LUC}$ estimates under transient carbon densities are derived by scaling their $E_{LUC}$ values with the average ratio of $E_{LUC}$ with transient densities to $E_{LUC}$ with static densities estimated from OSCAR and from BLUE. Scaling is done individually for each of the following $E_{LUC}$ subcomponents: total deforestation, total forest (re-)growth, gross sources from wood harvest, gross sinks from wood harvest, and other transitions. The resulting component-wise $E_{LUC}$ with transient densities estimates are then summed to obtain the net $E_{LUC}$ estimate for H&C23 and for LUCE. The uncertainty on $\delta L$ is estimated based on uncertainty estimates from BLUE and OSCAR. For BLUE, we estimate the $\delta L$ uncertainty (1 s.d.) across the estimates from the 7 DGVMs providing PFT-level output for soil and vegetation carbon[17]. For OSCAR, the $\delta L$ uncertainty is estimated as weighted standard deviation[18]. The $\delta L$ uncertainty for H&C23 and LUCE is derived as the average relative uncertainty of BLUE and OSCAR. The final $\delta L$ uncertainty is estimated using a random-effects model considering both the uncertainty estimates of each model and the variability of $\delta L$ estimates across bookkeeping models. The transient carbon densities correction ($\delta L$) leads to an increase in $E_{LUC}$ of $0.11 \pm 0.04$ GtC yr$^{-1}$ for the past decade.

### Land sink

**Replaced sinks and sources correction.** In the GCB, the natural land sink ($S_{LAND}$) is estimated using simulations from an ensemble of DGVMs that follow a common experimental protocol. Each model performs several simulations to isolate drivers of changes in land carbon fluxes. $S_{LAND}$ is estimated with the 'S2' simulation, where atmospheric $CO_2$ and climate vary over time, but land cover is held at pre-industrial (year 1700) levels. This set-up is designed to isolate the direct effects of increasing $CO_2$, climate change and nitrogen deposition on land carbon uptake, while excluding effects of direct human-driven land-use change. These latter are calculated separately in the $E_{LUC}$ flux estimated with the bookkeeping models. As land cover is fixed at pre-industrial levels, these S2 simulations represent the response of the land surface to increasing atmospheric $CO_2$, nitrogen deposition and changes in climate with too much forest cover globally (as forest area has decreased by about 20% since 1700). As carbon sinks in forests are typically larger than in other ecosystems, the $S_{LAND}$ term is overestimated. This issue is known as the replaced sinks and sources (RSS)[17,19] (in some publications also called the loss of sink capacity[21]). To address this issue, a recent study[48] developed a correction method that adjusts the $S_{LAND}$ estimate to reflect the actual historical land-cover distribution while still excluding carbon fluxes associated with direct human influences on land cover (for example, from deforestation, af/reforestation). The method uses a subset of seven DGVMs that simulate net biome production at the PFT level and include separate soil and litter carbon pools for each PFT. These models provide outputs from both the S2 simulation and the S3 simulation (varying $CO_2$, climate, and land use/cover). We extract the PFT-level net biome production from the S2 simulation and combine it with the time-varying land-cover fractions from S3. This allows us to reconstruct a corrected net biome production flux that reflects how the land system would respond to $CO_2$ and climate under the actual, changing land cover, while excluding anthropogenic land-use change emissions and sinks. We then compute the bias as the difference between the original $S_{LAND}$ (from the S2 simulation) and the reconstructed, land-cover-corrected $S_{LAND}$. The global correction is derived by summing grid-cell-level biases across the models, and the uncertainty is estimated from the inter-model standard deviation. This correction leads to a decrease of $S_{LAND}$ by $0.5 \pm 0.3$ GtC yr$^{-1}$ for the 2014–2023 period.

**Lateral carbon export correction.** In the GCB, the impact of human-induced changes in lateral carbon transfers on the land and ocean carbon sinks and $G_{ATM}$ have so far been excluded. Here we account for anthropogenic impacts on these lateral fluxes by taking the average of two recently published estimates: a data-ensemble method[24] and a process-based model that includes land-aquatic lateral exchanges and $CO_2$ fluxes with the atmosphere[25]. The two estimates are quantitatively consistent, are supported by a recent global assessment using another land surface model enabled for land-aquatic lateral exchanges (H. Zhang, personal communication) and are very close (within 10%), for their present-day carbon export estimate, to a recent global assessment relying on process-based models, observations and machine learning[49]. Extended Data Fig. 3 provides an overview of the different components of the carbon export correction. The anthropogenic perturbation (2014–2023 minus pre-industrial) on the lateral land-to-inland water carbon flux ($F'_{LI}$) amounts to $0.54 \pm 0.44$ GtC yr$^{-1}$ and is partitioned into increased aquatic $CO_2$ evasion ($F'_{IA}$, $0.34 \pm 0.26$ GtC yr$^{-1}$), aquatic carbon storage ($F'_{IS}$, $0.09 \pm 0.03$ GtC yr$^{-1}$) and carbon exports to the ocean ($F'_{IE}$, $0.11 \pm 0.08$ GtC yr$^{-1}$).

To estimate the impact of this enhanced lateral carbon export on $S_{LAND}$, we use the process-based estimate[25], which allows to separate the lateral land-to-inland water carbon flux ($F'_{LI}$) depending on the origin of the exported carbon. Incidentally, one half ($0.27 \pm 0.31$ GtC yr$^{-1}$) results from the transfer of dissolved $CO_2$ from the soil water column to the aquatic system, and the other half ($0.27 \pm 0.31$ GtC yr$^{-1}$) results from the transfer of terrestrial organic carbon to the aquatic system. The former (numbers in orange in Extended Data Fig. 3) represents a lateral displacement of $CO_2$ produced by soil heterotrophic respiration to the aquatic system ($F'_{IA}$, orange values), with no impact on the combined terrestrial + aquatic $CO_2$ flux to the atmosphere, and hence no impact on $S_{LAND}$. The latter (numbers in red in Extended Data Fig. 4) represents an additional loss from terrestrial ecosystems carbon reservoirs to the aquatic system, which can impact $S_{LAND}$. Indeed, out of the $0.27 \pm 0.22$ GtC yr$^{-1}$ of organic carbon lost from the terrestrial reservoirs, about one-quarter, $0.07 \pm 0.06$ GtC yr$^{-1}$, is transferred to inland waters, decomposed and released back to the atmosphere as $CO_2$, hence impacting $S_{LAND}$ ($F'_{IA}$, red values), whereas the remaining three-quarters are stored in other reservoirs ($0.09 \pm 0.03$ GtC yr$^{-1}$ buried in aquatic systems, $F'_{IS}$ and $0.11 \pm 0.08$ GtC yr$^{-1}$ exported to the open ocean, $F'_{IE}$), with no impact on $S_{LAND}$.

We do not correct the GCB estimate of the ocean sink ($S_{OCEAN}$), that is, we assume that the terrestrial carbon exported to the ocean ($F'_{IE}$, $0.11 \pm 0.08$ GtC yr$^{-1}$) remains stored in the ocean, as the fate of the land-derived carbon in the coastal and open ocean remains too uncertain to be quantified with confidence[24].

In summary, the LCE correction leads to a $0.07 \pm 0.06$ GtC yr$^{-1}$ reduction of $S_{LAND}$, with the uncertainty estimated by combining the

uncertainties reported in the original studies for enhanced $CO_2$ outgassing[24,25]. No LCE correction on $S_{OCEAN}$ was applied here.

## Ocean sink bias correction

In the GCB, the ocean carbon sink ($S_{OCEAN}$) is calculated as the mean of the ensemble average of GOBMs and the ensemble average of observation-based estimates ($fCO_2$ products). Both approaches are subject to known biases that are quantified here.

The evidence for the underestimation of the ocean $CO_2$ sink using GOBMs, already mentioned in GCB2024[1] comes from a number of studies, which all suggest an underestimation of around 10%. Comparison with interior ocean estimates of anthropogenic carbon accumulation suggests an underestimation of 8% (ref. 4) to 17% (ref. 33) for the periods 1994–2007 and 2004–2014, respectively. GOBMs produce a lower ocean sink compared with atmospheric inversions (by 16%) and atmospheric $O_2$-based estimates (by 24%), for the decade 2014–2023[1], although uncertainty ranges overlap. Process-based evaluation of the Earth system models also suggests a 9–11% underestimation of the ocean sink owing to biases in simulated Atlantic Meridional Overturning Circulation, Southern Ocean ventilation and surface-ocean Revelle factor[50], also qualitatively supported by regional studies[51-53]. A composite analysis of GOBMs and Earth system models suggests that GOBMs underestimate the ocean sink by 10% owing to inadequate spin-up strategies[34]. Regionally, eddy-covariance $CO_2$ flux data suggest a substantial underestimation of the Southern Ocean sink by the GOBMs[54]. All in all, although all lines of evidence have their own uncertainties, they consistently support that GOBMs underestimate the ocean sink. We thus have high confidence (90% confident) that the correction on the GOBMs estimate is positive. Hence, we propose a correction of +10% ± 8% based on the evidence provided above, with the uncertainty consistent with a 90% chance the correction is positive (Z-score = −1.28). The upwards scaling of the GOBMs by 10% results in an increase of the GOBM sink estimate by 0.26 ± 0.21 GtC $yr^{-1}$ for the 2014–2023 period.

Observation-based estimates ($fCO_2$ products) are built on direct measurements of the fugacity of $CO_2$ ($fCO_2$, which equals the partial pressure of $CO_2$ ($p_{CO_2}$) corrected for the non-ideal behaviour of the gas) from the Surface Ocean $CO_2$ Atlas (SOCAT)[26] that are gap filled using various statistical, regression and machine learning approaches. The air–sea $CO_2$ exchange is then calculated from the air–sea partial pressure difference of $CO_2$ and a wind-dependent bulk gas transfer formulation. These calculations do not consider temperature gradients arising from the surface warm layer and cool-skin effect (the less than 1-mm-thick surface micro-layer that cools through ocean heat loss to the atmosphere), which are mechanistically well understood but have historically been difficult to quantify. A recent study based on a field study of direct air–sea $CO_2$ fluxes suggests that the measurements need to be adjusted to consider a cool-skin effect (0.42 GtC $yr^{-1}$, increasing sink), which is in part offset by the effect of temperature differences between the measurement depth and the ocean surface (0.24 GtC $yr^{-1}$, decreasing sink), resulting in an upwards adjustment of the sink of 0.18 GtC $yr^{-1}$ (ref. 37). This is broadly consistent in magnitude with a GOBM model study that implemented the cool-skin effect[55]. For the cool-skin and warm-layer corrections of the $fCO_2$ products, the field study estimate comes without uncertainty[37]. However, based on the uncertainty estimate of the modelling study[55] and our expert judgement, we have medium confidence (66% confidence) that the correction is positive. Uncertainties remain, for example, owing to the lack of dedicated field campaigns and choice of rapid or equilibration model for the cool-skin correction[36,56], and should be resolved in the future to increase confidence. Hence, we propose a correction of 0.18 ± 0.4 GtC $yr^{-1}$, with the uncertainty consistent with a 66% chance the correction is positive (Z-score = −0.45). Additional warm bias leading to potential enhanced underestimation of the ocean sink has been identified also from variable sample depth and potential artificial

warming in the ship environment, but these factors are less well understood and constrained[35,36] and thus not further considered here.

In our revised assessment, we increase the GOBMs estimate by 10 ± 8% and the $fCO_2$ products estimate by 0.18 ± 0.4 GtC $yr^{-1}$. These two corrections combined lead to an increase of $S_{OCEAN}$ by 0.22 ± 0.23 GtC $yr^{-1}$ for the 2014–2023 period.

We note that the adjustment of both GOBM and $fCO_2$ product estimates does not resolve the discrepancy between them, but it does align the GCB mean ocean sink closer to independent estimates based on observations of the ocean interior and of atmospheric oxygen[3,4].

## Atmospheric $CO_2$ growth rate estimate

In the GCB, the global atmospheric $CO_2$ annual growth rate is derived from $CO_2$ mole fraction observations at the surface (in ppm $yr^{-1}$), which are converted to mass growth rates ($G_{ATM}$, in GtC $yr^{-1}$) using a conversion factor with a constant value of 2.124 GtC $ppm^{-1}$ (ref. 40). Here we estimate the uncertainty in the conversion factor and hence $G_{ATM}$, using the 14 atmospheric inversions included in GCB2024, following the method by ref. 57. We use the model-sampled mole fractions at the surface stations to calculate the annual $CO_2$ growth rate (in ppm $yr^{-1}$), following the same calculation for the observations as developed by ref. 41, similar to the method used by the National Oceanic and Atmospheric Administration[39]. We calculate the annual net input of $CO_2$ in the atmosphere (in GtC $yr^{-1}$) as the sum of the annual fossil-fuel emissions and the inverse-derived net land and ocean sinks. The annual ratio of this net annual input of $CO_2$ divided by the annual growth rate gives the conversion factor (in GtC $ppm^{-1}$). This is repeated for each inverse model and results in annual estimates of the conversion factor (Extended Data Fig. 4a), with their standard deviation. It is noted that not all inversions are available over the complete period, and we therefore focus the analysis on the period covered by most inversions (2001–2023). The conversion factor shows statistically significant interannual variability that is larger than the standard deviation of the 14 inverse models (Extended Data Fig. 4a). We subsequently propagate the uncertainty in the conversion factor resulting from (1) the annual uncertainty in the observation-based growth rate, (2) the mean interannual variability over the 2001–2023 period and (3) the mean standard deviation of the inversions over 2001–2023, to estimate the resulting uncertainty on $G_{ATM}$ (in GtC $yr^{-1}$) (Extended Data Fig. 4b). Finally, we propagate this combined uncertainty to the GCB $B_{IM}$, where the uncertainty band represents the uncertainty in the $B_{IM}$ explained by the $G_{ATM}$ uncertainty (Extended Data Fig. 4c). Years within this uncertainty band therefore do not have a statistically significant $B_{IM}$. No adjustment on $G_{ATM}$ itself is made here as the year-to-year changes in the conversion factor need further evaluation.

## Climate change impact on the GCB

The land and ocean sinks in the GCB account for both the effect of increasing atmospheric $CO_2$ and climate change over the historical period. As described in GCB2024, the DGVMs and GOBMs performed two simulations: one accounting for changes in atmospheric $CO_2$ and climate, and one with the same prescribed increase in atmospheric $CO_2$, but with a constant climate forcing, representative of a natural climate (1900–1910 for the DGVMs, late 1950s for the GOBMs). The difference between these two simulations is the effect of climate change on the land and ocean sinks ($S_{LAND}^{clim}$, $S_{OCEAN}^{clim}$), as simulated by the DGVMs and GOBMs (Fig. 2 and Extended Data Fig. 5). We add these climate change effects on the revised estimates of $S_{LAND}$ and $S_{OCEAN}$ to estimate the land and ocean sinks in the absence of climate change. The impact on atmospheric $CO_2$ (Fig. 2c) is estimated as $G_{ATM}^{clim} = AF \times (S_{LAND}^{clim} + S_{OCEAN}^{clim})$, where AF is the airborne fraction. The theoretical atmospheric $CO_2$ growth rate, in the absence of climate change, is then estimated as $G_{ATM} - G_{ATM}^{clim}$.

## Data availability

All data presented in this paper are available via Zenodo at https://zenodo.org/records/16367993 (ref. 58).

## Code availability

No new code was generated for this study. Figures with maps were done using the R statistical environment.

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

**Acknowledgements** We thank the Global Carbon Project, R. Jackson and all contributors to annual updates of the GCB, which have informed continuous assessments of our understanding of the carbon cycle and its evolution under human pressure. This work received support through Schmidt Sciences, LLC (VESRI CALIPSO project and OBVI InMOS project). C.L.Q.was supported by the UK Royal Society (RSRP\R\241002). M.W.J. was funded by the Natural Environment Research Council (NE/V01417X/1). G.P.P. and R.M.A. were supported by the Research Council of Norway (NorSink, 352474). A.v.d.W., W.P. and I.T.L. received funding from the Netherlands Organisation for Scientific Research (grant number NWO-2023.003 and VI.Vidi.213.143). H.T. acknowledges support from US Department of Agriculture (TENX12899). J.G.C. was supported by Australia's NESP2-Climate Systems Hub. J.H. acknowledges funding from ERC-2022-STG OceanPeak (grant number 101077209) (European Commission). The work reflects only the authors' view; the European Commission and their executive agency are not responsible for any use that may be made. For the purpose of open access, the author has applied a Creative Commons Attribution (CC BY) licence to any author accepted manuscript version arising from this submission.

**Author contributions** P.F. and C.L.Q. designed the study and drafted the paper. J.P., C.S. and T.G. provided the revised land-use-emissions estimate. M.O'S., S.S., P.R. and H.T. provided the revised land sink estimate. J.H., P.L., D.C.E.B., A.O. and C.L.Q. provided the revised ocean sink estimate. I.T.L., W.P., A.v.d.W., X.L., E.M. and H.L. assessed the variability and uncertainty in the atmospheric concentration growth rate. R.M.A. and G.P.P. assessed the fossil-fuel emissions estimate. M.W.J., J.G.C. and P.C. commented on the draft. All authors contributed to the writing of the paper.

**Competing interests** The authors declare no competing interests.

**Additional information**
**Correspondence and requests for materials** should be addressed to Pierre Friedlingstein.

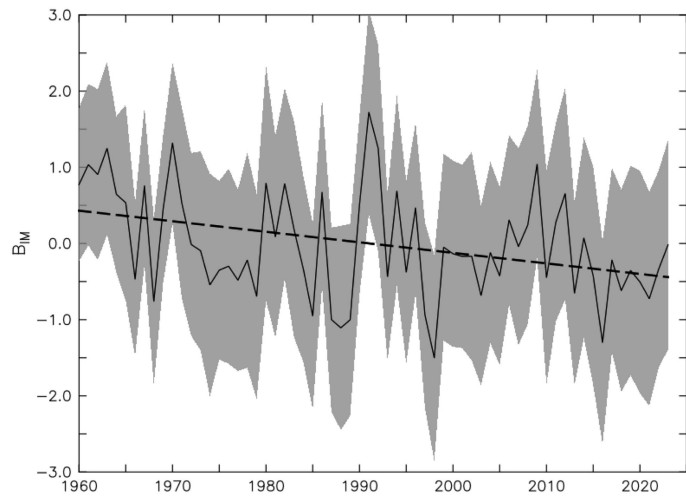

**Extended Data Fig. 1 | Budget imbalance.** ($B_{IM}$) as reported in the GCB2024, as reported in the GCB2024, showing a statistically significant negative trend (dotted line) of $-0.14 \pm 0.04$ GtC/yr per decade (p-value = 0.003). Units are GtC/yr.

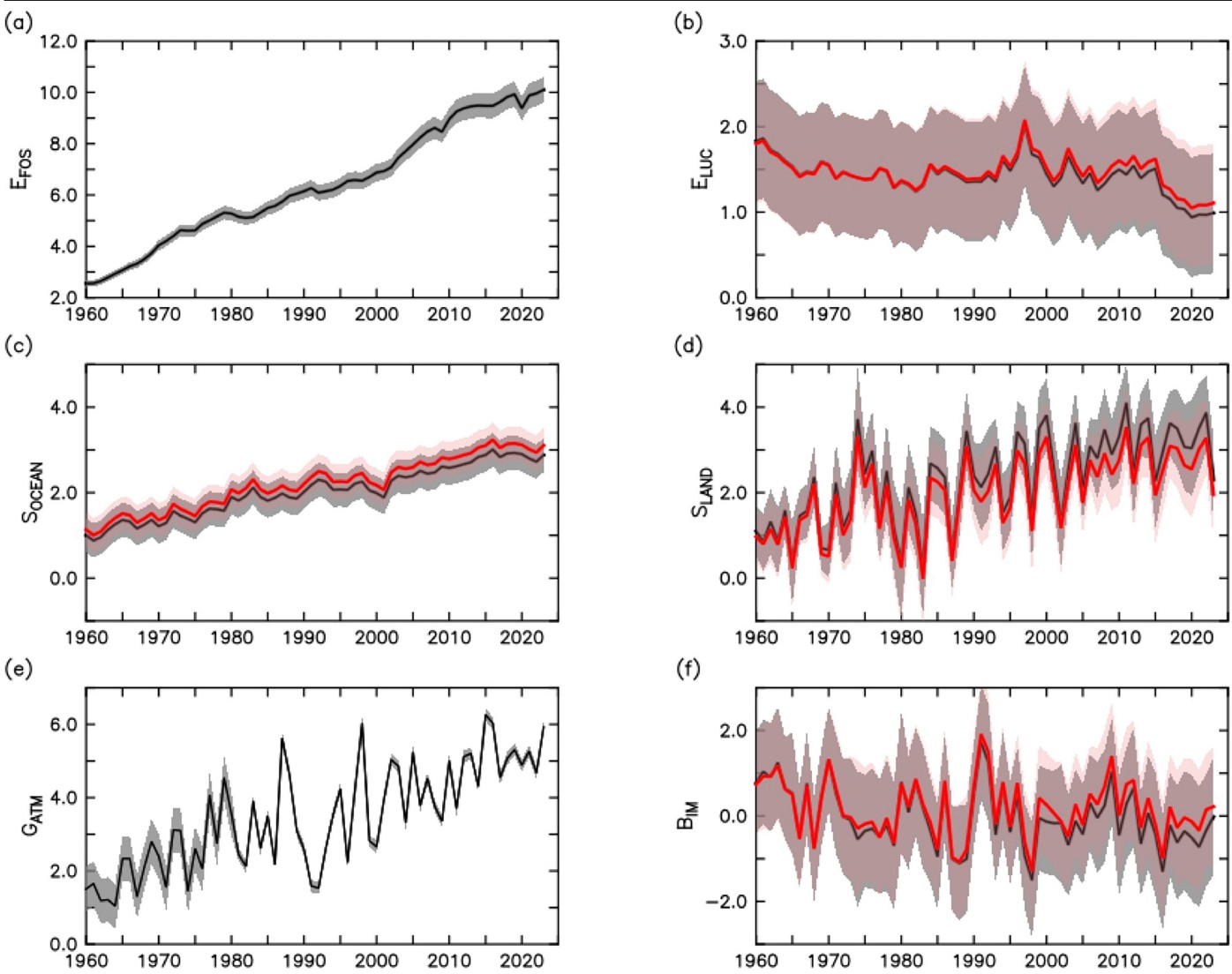

**Extended Data Fig. 2 | Consolidated global carbon budget.** Revision (in red) compared to the GCB2024 estimate (in black) of (b) net land-use emissions, (c) ocean sink, (d) land sink, and (f) budget imbalance. Panels (a) fossil CO2 emissions and (e) atmospheric CO2 growth rate are unchanged. All fluxes are in GtC/yr.

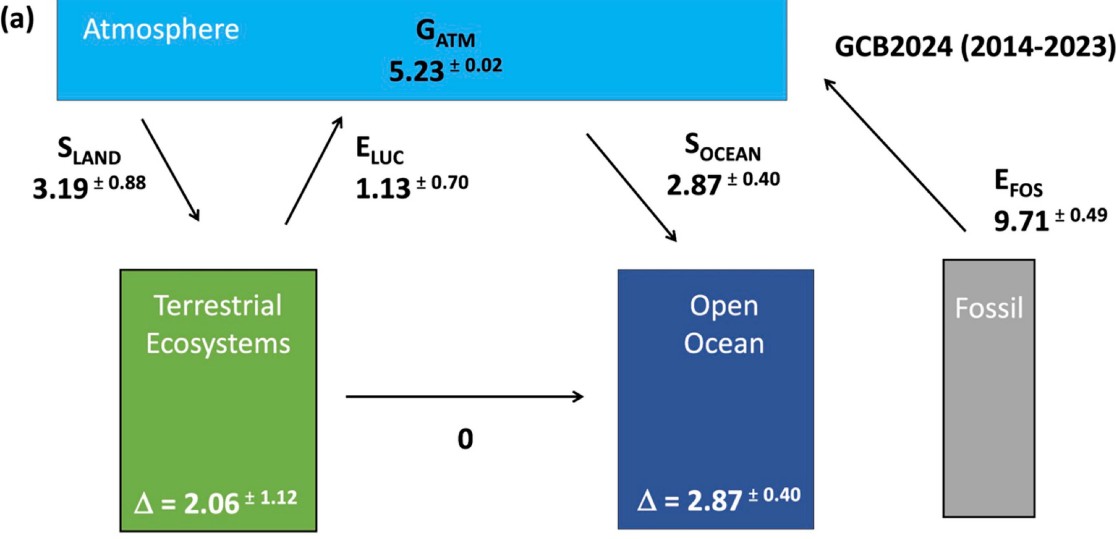

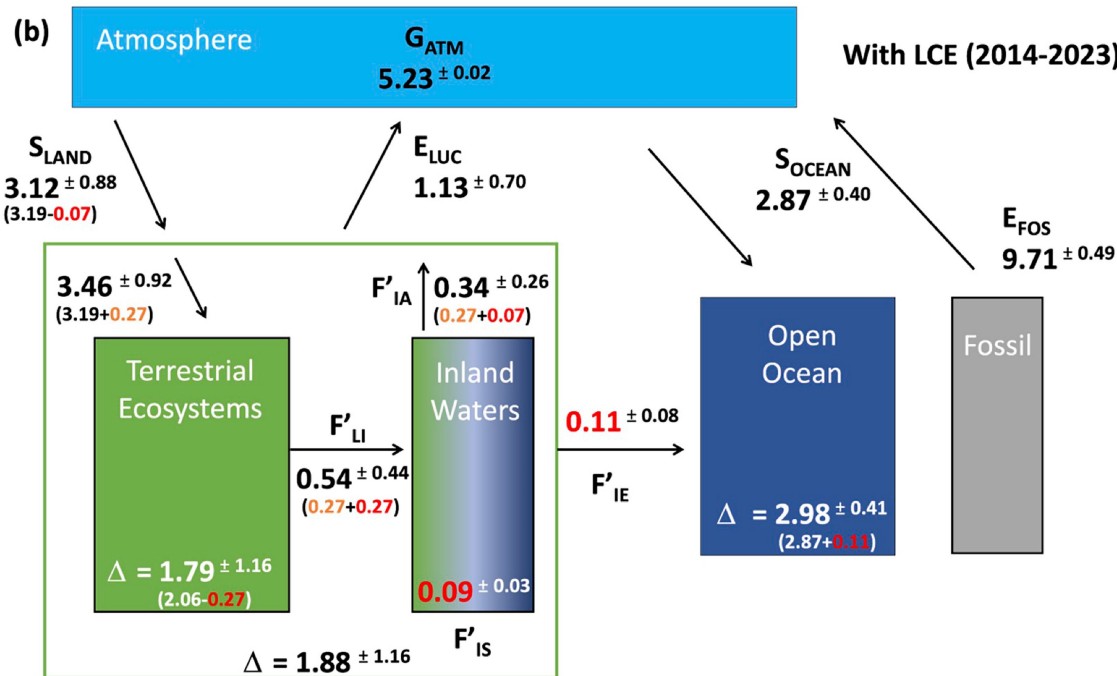

**Extented Data Fig. 3 | Impact of lateral carbon flux correction on SLAND.**
Global carbon budget (2014-2023) without (a) and with (b) historical changes in lateral carbon fluxes. Units are GtC/yr. The additional green/blue box represents inland waters, and the surrounding green open rectangle represents the whole land system (terrestrial ecosystems and inland waters combined). The perturbations on inland water fluxes follow the nomenclature of ref. 24 and represent land-to-inland water flux ($F'_{LI}$), aquatic $CO_2$ outgassing ($F'_{IA}$), aquatic

carbon storage ($F'_{IS}$) and lateral carbon exports to ocean ($F'_{IE}$). All fluxes were quantified as the mean of values reported by refs. 24,25 and Zhang, pers com. $F'_{IA}$ is subdivided into contributions from soil-derived $CO_2$ (in orange) and $CO_2$ from soil organic carbon (in red) respired in inland waters. The Δ represents changes in carbon storage in the different reservoirs. The net effect on $S_{LAND}$ is a decrease of 0.07 ± 0.06 GtC/yr. See methods for further details.

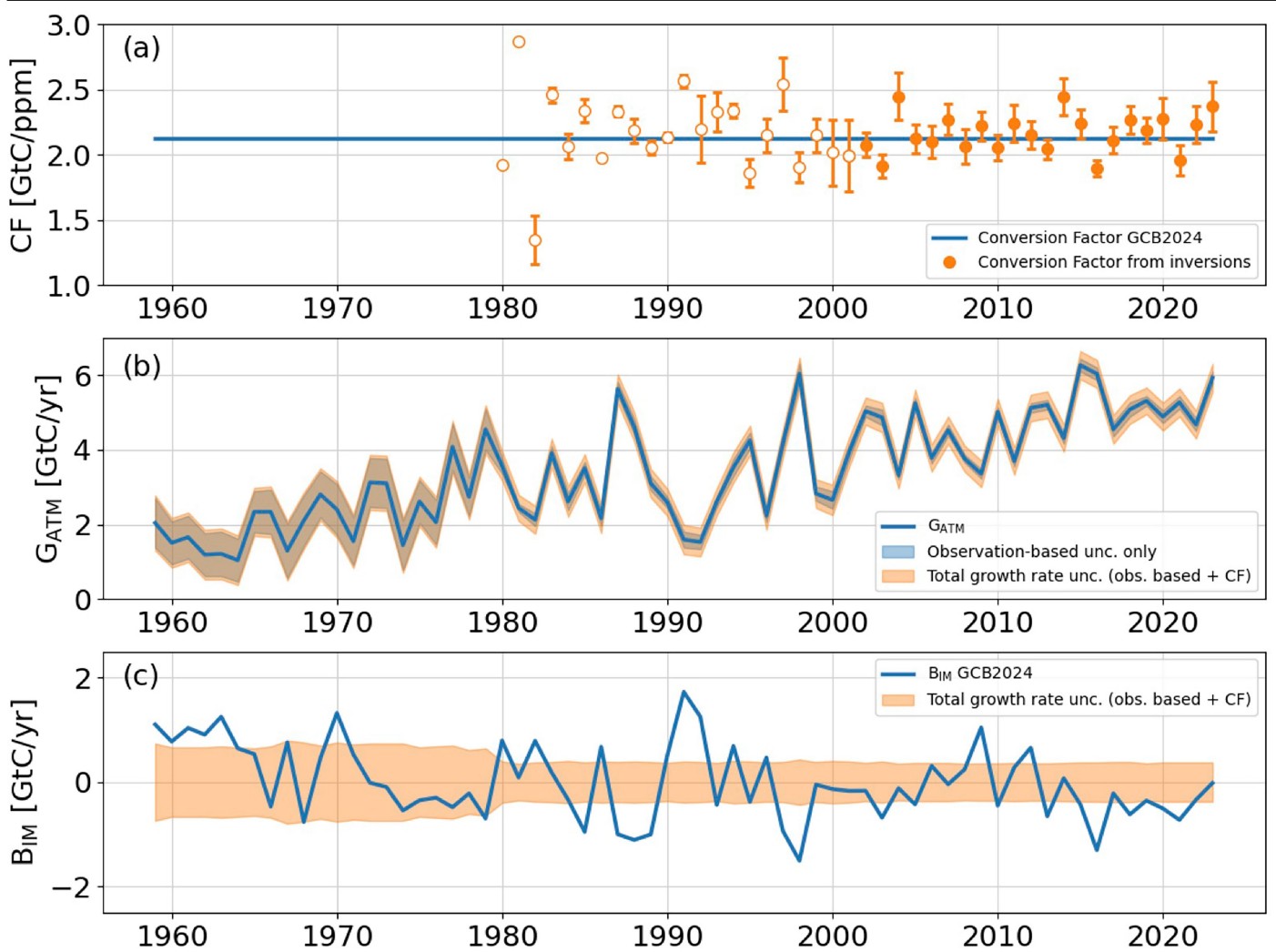

**Extended Data Fig. 4 | Atmospheric growth rate.** Annual conversion factors (CF, in GtC/ppm) for converting the observation-based atmospheric growth rate [ppm/yr] to atmospheric mass growth rates [GtC/yr] derived from the 14 atmospheric inversions included in GCB2024 (orange) in comparison to the fixed value currently used in GCB2024 (blue), open symbols represent years in which less than 4 atmospheric inversions are available; (b) atmospheric growth rate ($G_{ATM}$) with propagated uncertainty from: 1) uncertainty in the annual observation-based growth rate [ppm/yr], shown in blue shading, 2) mean interannual variability in the CF over 2001-2023, and 3) mean standard deviation of the inverse CFs over 2001-2023 (total combined uncertainty shown in orange shading); and (c) the GCB2024 budget imbalance ($B_{IM}$) [GtC/yr] with the propagated uncertainty in $G_{ATM}$ (orange shading).

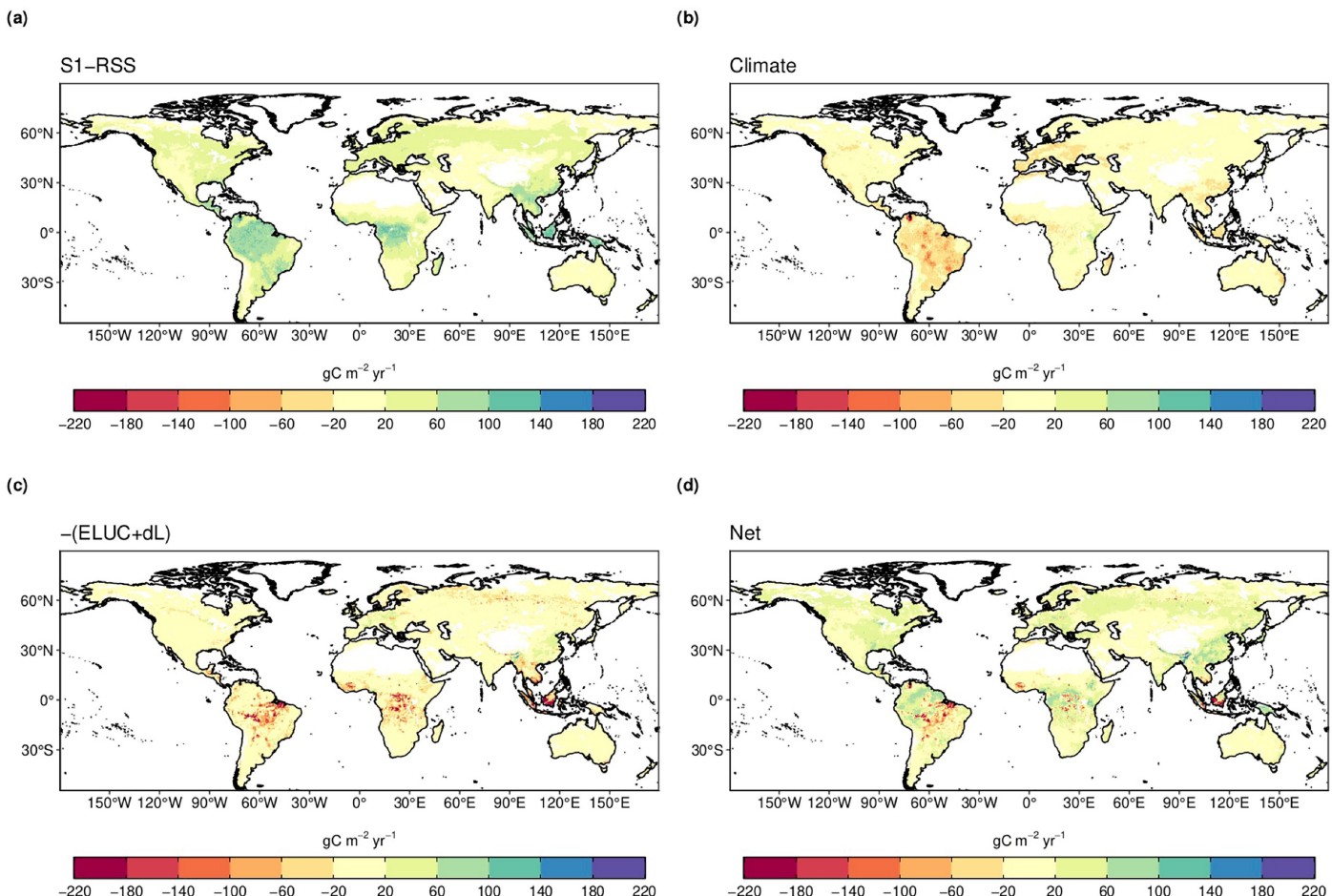

**Extended Data Fig. 5 | Land CO₂ fluxes.** (a) Land carbon sink due to atmospheric CO₂ increase (CO₂ fertilization) only, (b) effect of climate change on the land carbon flux, (c) land carbon flux due to land-use change, (d) net land CO₂ flux (a + b + c). Positive values indicate sinks, negative values indicate sources. Units are gC/m²/yr.

**Extended Data Table 1 | Decadal average of all components of the consolidated global carbon budget (GtC/yr)**

| | $G_{ATM}$ | $E_{FOS}$ | $E_{LUC}$ | $S_{LAND}$ | Net Land | $S_{OCEAN}$ | $B_{IM}$ |
|---|---|---|---|---|---|---|---|
| GCB2024 | 5.2±0.02 | 9.7±0.5 | 1.1±0.7 | 3.2±0.9 | 2.1±1.1 | 2.9±0.4 | -0.4±1.3 |
| Revised $E_{LUC}$ δL | | | 1.2±0.7 (+0.1±0.04) | 3.2±0.9 | 2.0±1.1 | | -0.3±1.3 |
| Revised $S_{LAND}$ RSS | | | | 2.75±0.9 (-0.46±0.3) | 1.5±1.1 | | 0.1±1.3 |
| Revised $S_{LAND}$ LCE | | | | 2.7±0.9 (-0.07±0.06) | 1.4±1.1 | | 0.2±1.3 |
| Revised $S_{OCEAN}$ | | | | | | 3.1±0.5 (+0.22±0.23) | -0.02±1.3 |
| **This Study** | **5.2±0.02** | **9.7±0.5** | **1.2±0.7** | **2.7±0.9** | **1.4±1.1** | **3.1±0.5** | **-0.02±1.3** |
| Atmospheric inversions | 5.2±0.0 | 9.7±0.5 | N/A | N/A | 1.4±0.5 | 3.1±0.5 | 0 |
| Atmospheric oxygen | 5.2±0.0 | 9.7±0.5 | N/A | N/A | 1.0±0.8 | 3.4±0.5 | 0 |

*Net Land* is the net land $CO_2$ flux, calculated as $S_{LAND}$ - $E_{LUC}$. Atmospheric inversions and atmospheric oxygen do provide *Net Land* but do not separate $E_{LUC}$ from $S_{LAND}$. The budget imbalance ($B_{IM}$) is the difference between anthropogenic net emissions ($E_{FOS}$+$E_{LUC}$) and accumulation of carbon in the atmosphere, land and ocean ($G_{ATM}$+$S_{LAND}$+$S_{OCEAN}$). By design, atmospheric inversions and atmospheric oxygen budget imbalance is null.

**Extended Data Table 2 | Decadal average of all components of the consolidated global carbon budget (GtC/yr)**

|  |  | 1960s | 1970s | 1980s | 1990s | 2000s | 2014-2023 |
|---|---|---|---|---|---|---|---|
| Net emissions | $E_{FOS}$ | 3.0±0.2 | 4.7±0.2 | 5.5±0.3 | 6.4±0.3 | 7.8±0.4 | 9.7±0.5 |
|  | $E_{LUC}$ | 1.6±0.7 | 1.4±0.7 | 1.4±0.7 | 1.6±0.7 | 1.5±0.7 | 1.2±0.7 |
| Partitioning | $G_{ATM}$ | 1.7±0.07 | 2.8±0.07 | 3.4±0.02 | 3.1±0.02 | 4.0±0.02 | 5.2±0.02 |
|  | $S_{OCEAN}$ | 1.3±0.5 | 1.6±0.5 | 2.1±0.5 | 2.3±0.5 | 2.5±0.5 | 3.1±0.5 |
|  | $S_{LAND}$ | 1.0±0.5 | 1.7±0.8 | 1.5±0.8 | 2.0±0.6 | 2.4±0.7 | 2.7±0.9 |
|  | $B_{IM}$ | 0.5±1.0 | 0.1±1.2 | -0.02±1.2 | 0.4±1.2 | 0.3±1.1 | -0.1±1.3 |