## [Peer Review File · Nature]

Emerging climate impact on carbon sinks in a consolidated carbon budget

Corresponding Author: Professor Pierre Friedlingstein

Note from the editor: In total, there were four referees of this manuscript, labelled as reviewers #1, #2, #4, and #5.

Version 0:

Reviewer comments:

Referee #1

(Remarks to the Author)

Review of Friedlingstein et al 2025, submitted to Nature

Friedlingstein and colleagues present evidence to revise the approaches used in the Global Carbon Budget, which is released annually. One can expect that the adjustments to the budget proposed in this paper will be incorporated into the Global Carbon Budget going forward, making it an important contribution. However, it should be noted that the paper does not present new data, models, or analyses; instead, it summarizes many other studies and connects them within the context of the global carbon budget.

It is also important to note that the adjustments to the budget proposed are not statistically significantly different from the GCB2024 (Friedlingstein et al. 2025). Table 1 shows that while mean values have shifted, no changes are outside the previously-reported 1-sigma uncertainties. This means that there is no new understanding of the global carbon budget here. We encourage the authors to be more transparent on this issue, as it points to the lack of progress on uncertainties and the need for improved observation and modeling.

Major Comments

1. Our primary concern with the paper is the treatment of uncertainty.

a. At the very top level, one notes from Table 1 that the revisions to the Global Carbon Budget proposed here are not statistically significant. Mean values have changed, but none of these adjustments are outside the previously-reported 1-sigma uncertainties.

i. The authors need to make it clear to the reader that these adjustments to the Global Carbon Budget are not statistically significant. This should be stated in the Abstract and in the Implications sections, at minimum.

ii. That the BIM still has large uncertainty is made clear in this version of the paper, but the authors also need to discuss the fact that these "changes in the budget" are also not significant.

b. All of the bar plots in the paper should include uncertainty bars. At present, these are found only on Figure 3 and 4. They are needed on Figure 1 and S6.

i. Where figures are labeled with quantitative values, uncertainties should be included. These figures have lots of white space of which to make use.

c. All numbers in the text should include uncertainty. Only some do in this version.

2. A second concern is the lack of clarity in the Methods regarding the actual adjustments made. The authors state justifications for adjustments that imply the approach taken, but do not consistently make clear statements about precisely what was done, e.g. "GOBM air-sea fluxes as reported were increased by 10%". We suggest a short summary at the end of each Supplementary section to explicitly state the adjustments that are made for each component of the budget.

Minor comments

• We note that the authors neglect to include page and line numbers in the manuscript, adding a burden for Reviewers. We use the headers of the sections here. The authors are requested to please include page and line numbers in future versions

of their manuscript.

- When describing results, please refer to specific figure panels (e.g., Fig. 2a) when appropriate.

Abstract

- “continue” not “continued”
- “sinks, contributing”
- Please include an explicit statement that the revisions to the mean values of the budget terms are not statistically significantly different from previous estimates given their 1-sigma uncertainties.

Introducing latest evidence

- Please explain why the authors do not revise to the pre-industrial land-to-ocean export of 0.55 ± 0.1 PgC/yr published by Liu et al. 2024 in Nature Geoscience.
- Cite uncertainty on 0.34 GtC/yr
- It is not clear what is meant by “uniquely”
- Cite uncertainty on 3.2 GtC/yr 2.6 GtC/yr
- Add reference after “atmospheric carbon and oxygen” to Keeling paper
- Please explain ‘statistical differences’ - the term and the use of quotations
- Strike “allow to determine” and replace with “supporting determination of”

Revised estimates of the global carbon budget

- “(p-value = 0.003; Supplementary Fig. 3)”

Land use change correction

- CO₂, not CO₂

Land sink replaced sinks and sources correction (RRS)

- Consider re-writing the header
- Consider re-writing this sentence: “This issue has been referred to as the loss of sink capacity is called here the replaced sinks and sources (RRS)”

Land sink lateral carbon export correction

- For consistency, use either “machine-learning” or “machine learning”

Atmospheric CO₂ growth rate from surface concentrations

- “This results in inverse-based annual estimates of the CF (Supplementary Fig. 5c)”: Should this be 5a and not 5c?

Airborne fraction

- This is the first time Supplementary Figure 2 is mentioned. Perhaps move/re-number this figure
- BIM, not BIM

Climate change impact on the global carbon budget

- CO₂, not CO₂

Methods

Land use Change correction

- Define TRENDYS2 / TRENDY S2
- State explicitly the adjustments made

Land Sink replaced...

- Strike “as we have shown”
- State explicitly the adjustments made

Land sink lateral ..

- Replace “largely relying on observations and machine learning” with “relying on process models, observations and machine learning”
- Include uncertainty on all numbers
- State the time frame of analysis
- Explain better what justifies the choice to not include the 0.11 GtC adjustment in the ocean, i.e. justify the choice to assume this carbon is stored
- State explicitly the adjustments made

Ocean Sink

- Include uncertainty on all numbers
- State explicitly the adjustments made
- Add a reference for the sentence beginning “Correcting the GOBM...”

Atmospheric CO₂...

- Explain in a few sentences the method of van der Woude et al. Readers should be able to understand this to first-order without having to read another paper.
- State explicitly that no adjustments have been made

Figure 1 is difficult to understand, and it is difficult to see the adjustments being proposed.

- Uncertainty bounds needed
- Colors should be consistent with Figure 3 for each term
- The Atmosphere and FF terms should be indicated somehow; explain abbreviations in the figure caption (dL, RSS, LCE)
- Please include subpanel labeling and refer to these in text
- Please increase font size
- There is a lot of white space here that can be used to improve this figure; one of the most interesting aspects about this figure is the difference between GCB2024 and this study, but these differences are very difficult to see
- Add a, b, c, d
- Add GtC/yr label
- Consider combining this figure with Figure 3. This could show GCB2024 in top row, adjustments (with reduced vertical extent) in middle, and final in bottom. All with uncertainty bounds. This will help the reader more clearly understand the adjustments being made.

Table 1/S1

- There is a lot of white space here that can be used to improve these tables. It would also be useful to add a row that shows the difference between GCB2024 and this study. We suggest something like: (see attached Picture1.png)

Figure 3

- Use the appropriate subscript for the x-axis labels (EFOS, ELUC etc.)
- Indicate which components have been updated in this study; e.g., bold x-axis label, or use thicker box edge lines
- Change figure description to: "Revised Global Carbon Budget"
- It would be useful to compare to GCB2024; could you add boxes for the GCB2024 in the background?
- As noted above, consider combining with Figure 1 to make a single figure

Figure S2

- Remove white space

Figure S4

- Caption needs to indicate what gray box is, and what the dates are
- The boxes need to be labeled (ATM, Ocean, etc)
- (b) has 0.09 and 0.11 switched from where they should be based on the text
- Increase font size
- Consistently use the same labels (Fie, FLI, etc.) as in the caption and text on the figure.
- Include uncertainties on all numbers

Figure S6

- Uncertainty bounds are needed

Figure S7

- Remove white space

Figure S8

- CO₂, not CO₂

Figure S9

- Please label the lines with a legend
- The text suggests that the blue includes the impact of climate change, but the caption says this study is black. Both numbers appear to derive from this study. Please clarify.

Referee #2

(Remarks to the Author)

I co-reviewed this manuscript with one of the reviewers who provided the listed reports.

Referee #4

(Remarks to the Author)

Title: Trends in sources and sinks of carbon dioxide over the past 65 years.

Summary:

Here the authors make minor adjustments to the Global Carbon Budget from 2024 (GCB2024), by altering land use emissions and how they are treated in models, that appear to have a big impact in reducing the net land C sink. The question is whether this study really advances our understanding of the global carbon budget (beyond that of GCB2024) or just found a way to explain the residual variance not predicted by land models. We are not convinced given the uncertainty of land use change fluxes and their potential impact on questionable trends in the airborne fraction. Furthermore, given these uncertainties in the global fluxes, updates to regional fluxes may be less credible, or at least need further discussion.

General Comments:

In general, land use change (LUC) fluxes have the greatest relative uncertainty in the carbon budget (~50%) and therefore any changes in land use and how they are treated can have large impacts on inferred trends in the global land C sink. Land use changes can lead to changes in CO₂ two-way fluxes-not just one-way emissions. The authors here assume that all land use changes lead to a positive emission of CO₂ to the atmosphere; however, there are instances where LUC over time can lead to negative CO₂ fluxes from the atmosphere. For instance, historic land use change over the central North America undoubtedly lead to a release of CO₂ to the atmosphere when primary forests were converted to grazing and farming lands. However, the intensive agricultural land use occurring today over central North America explains most of the continental net land CO₂ sink (Liu et al., 2018). Thus land use over parts of North America appears to have transformed ecosystems from CO₂ sources and then back to CO₂ sinks over time.

Because there is so much uncertainty associated with mean annual land use fluxes any trends in land use fluxes over time should be interpreted with caution. For instance, these very same co-authors have previously reported apparent increasing trends in land use fluxes (Hong et al., 2021), in contrast to the decreasing trends reported here (Fig. S3, regardless of C density). Furthermore, previous analyses have investigated how slight changes in land use emissions associated with deforestation in particular can affect our estimates of the land and ocean C sinks over time as indicated by the airborne fraction (van Marle et al., 2022) and this paper was retracted due to dubious statistics. Based on this previous retraction, it is surprising that the authors appear to rely on simple correlations of the airborne fraction over time, rather than more robust Mann-Kendall trend tests or Monte-Carlo simulations (Bennedsen et al., 2023). Lastly, the authors selectively cite previous studies reporting significant trends in the airborne fraction that are somewhat dated and dubious at best. More recent

detailed analyses of the airborne fraction show no significant trend and considerable variability in apparent trends depending upon the land use change emissions considered (Bennett et al., 2024).

The way in which the current results are presented makes it unclear whether the main conclusion is that the land carbon sink has always been overestimated, or that the land sink is starting to saturate and taking up less carbon relative to the ocean. Although there are subtle differences in these conclusions, the implications of these different conclusions are quite substantial and not really distinguished in the conclusions. For instance, the conclusion in the abstract that 'The combined effects of climate change and deforestation turns South American and Southeast Asian ecosystems from CO₂ sinks to sources.' - seems to be at direct odds with recently updated forest inventory data that show a persistent land sink due in part to increases in forest plot density and tropical forest regrowth (Pan et al., 2024). Is secondary forest regrowth considered at all in these revised land use change flux estimates? Lastly, if the argument is that the land portion of the global carbon sink has diminished over time, then there are several recent satellite records that may corroborate this finding (Li et al., 2023; Wang et al., 2020) and could be used to put these results into context, especially at the regional scale. However, these studies were not cited.

The regional characteristics of LUC flux and its contribution to the updated global carbon budget need better explanation. Figure 3 displayed the LUC flux at a global scale, and Figure 4 displayed regional contributions. However, It would be good to see how updated LUC fluxes affect the revised global and regional carbon budgets, and which region(s) dominate the budget difference between GCB2024 and the revised GCB. It is possible a few particular regions dominate the revised GCB and budget balance.

The increase in LUC flux in this revised GCB estimate appears to be largely due to the use of transient carbon density, rather than the land use data itself. In this case, the transient model estimated a higher transient carbon density than static carbon density in GCB 2024, such that there is higher carbon emission per unit area in land use change. Correct? If so, why is there a higher transient carbon density? Is this density of the biomass (gC/m³) or is it ecosystem density (gC/m²) as estimated by Pan et al. 2024 that is necessary to solve for the persistent forest C sink? Is this due to changing environmental factors, such as atmospheric CO₂ and climate, or does the dynamic global vegetation modeling capture higher growth? Furthermore, is this transient carbon density incorporated into the ensemble of models considering the replaced sinks and sources (RSS) and if so doesn't this suggest that 'land use emissions' are just a subset of the net land flux term?

Specific Comments (see comments in PDF):

The title is not very compelling and belies the abstract- maybe something revealing the discovery here.

Be consistent with units. Some figures use PgC (preferred) and others GtC (less preferred).

References:

- Bennedsen, M., Hillebrand, E., & Koopman, S. J. (2023). On the evidence of a trend in the CO₂ airborne fraction. *Nature*, 616(7956), E1–E3. <https://doi.org/10.1038/s41586-023-05871-6>
- Bennett, B. F., Salawitch, R. J., McBride, L. A., Hope, A. P., & Tribett, W. R. (2024). Quantification of the airborne fraction of atmospheric CO₂ reveals stability in global carbon sinks over the past six decades. *Journal of Geophysical Research. Biogeosciences*, 129(3). <https://doi.org/10.1029/2023jg007760>
- Hong, C., Burney, J. A., Pongratz, J., Nabel, J. E. M. S., Mueller, N. D., Jackson, R. B., & Davis, S. J. (2021). Global and regional drivers of land-use emissions in 1961–2017. *Nature*, 589(7843), 554–561. <https://doi.org/10.1038/s41586-020-03138-y>
- Li, F., Xiao, J., Chen, J., Ballantyne, A., Jin, K., Li, B., Abraha, M., & John, R. (2023). Global water use efficiency saturation due to increased vapor pressure deficit. *Science*, 381(6658), 672–677. <https://doi.org/10.1126/science.adf5041>
- Liu, Z., Ballantyne, A. P., Poulter, B., Anderegg, W. R. L., Li, W., Bastos, A., & Ciais, P. (2018). Precipitation thresholds regulate net carbon exchange at the continental scale. *Nature Communications*, 9(1), 3596. <https://doi.org/10.1038/s41467-018-05948-1>
- Pan, Y., Birdsey, R. A., Phillips, O. L., Houghton, R. A., Fang, J., Kauppi, P. E., Keith, H., Kurz, W. A., Ito, A., Lewis, S. L., Nabuurs, G.-J., Shvidenko, A., Hashimoto, S., Lerink, B., Schepaschenko, D., Castanho, A., & Murdiyarso, D. (2024). The enduring world forest carbon sink. *Nature*, 631(8021), 563–569. <https://doi.org/10.1038/s41586-024-07602-x>
- van Marle, M. J. E., van Wees, D., Houghton, R. A., Field, R. D., Verbesselt, J., & van der Werf, G. R. (2022). New land-use-change emissions indicate a declining CO₂ airborne fraction. *Nature*, 603(7901), 450–454. <https://doi.org/10.1038/s41586-021-04376-4>
- Wang, S., Zhang, Y., Ju, W., Chen, J. M., Ciais, P., Cescatti, A., Sardans, J., Janssens, I. A., Wu, M., Berry, J. A., Campbell, E., Fernández-Martínez, M., Alkama, R., Sitch, S., Friedlingstein, P., Smith, W. K., Yuan, W., He, W., Lombardozzi, D., ... Peñuelas, J. (2020). Recent global decline of CO₂ fertilization effects on vegetation photosynthesis. *Science*, 370(6522), 1295–1300. <https://doi.org/10.1126/science.abb7772>

Referee #5

(Remarks to the Author)

I co-reviewed this manuscript with one of the reviewers who provided the listed reports.

Version 1:

Reviewer comments:

Referee #1

(Remarks to the Author)

Based on a variety of recent evidence, Friedlingstein and colleagues suggest revisions to the datasets supporting the annual Global Carbon Budget. With these adjustments applied, the mean of the Budget Imbalance is reduced to near zero. As these adjustments to the budget can be expected to be directly incorporated into the Global Carbon Budget going forward, it is an important contribution.

We have reviewed the revised manuscript and find that the authors have adequately addressed all our comments from the first round of review. We recommend publication.

Referee #2

(Remarks to the Author)

I co-reviewed this manuscript with one of the reviewers who provided the listed reports.

Referee #4

(Remarks to the Author)

Friedlingstein et al. Second Review

We appreciate the authors revisions to the previous draft of this paper and the depth of their knowledge on this topic. However, we still have concerns whether their revised global carbon budget advances our insight into global carbon cycle processes, or simply reconciles model simulations with observations.

Much of their analysis has focused on reconciling the apparent imbalance between the inferred land C sink and the model simulated land C sink, which is a persistent problem that has vexed C cycle scientists for some time. In fact, in their response they state-

'The fact that the statistically significant trend, previously present in the carbon budget imbalance, is now completely resolved and indistinguishable from zero is a major advance of our manuscript.'

I agree that this is a major contribution; however, I would also encourage them to be a bit more cautious. They are concluding that they have been able to perfectly match theory (ie. models) with observations (ie. atmospheric/ocean inversions). If this is true then there is no need for further terrestrial model development, but this is not true because we know that terrestrial models are still missing critical processes, such as non-respiratory C loss pathways and lateral C transport. Furthermore recent analyses have even questioned whether the inferred C land sink is overestimated and propose other potential solutions to this problem (Randerson et al., 2025). Perhaps a statement at the conclusion of the paper noting that they have been able to reconcile the 'net land C sink' but that future model development is necessary to advance our understanding of the underlying processes.

The authors acknowledge that land use change fluxes can be positive or negative. In fact, one of the authors has a paper on the uncertainty of land-use fluxes (Pongratz et al. 2014) and thus referring to them strictly as emissions is misleading:

'parts of the world including North America, Europe, and China are currently net carbon sinks from land-use change (see Figure 6b of GCB2024, <https://doi.org/10.5194/essd-17-965-2025>), mainly due to forest regrowth. However, for decades and when assessed on the global scale, the net effect of anthropogenic land-use change has been a source to the Atmosphere.'

The authors are following the standard in the literature, but when the standard is misleading and cause estimates to be biased towards positive fluxes to the atmosphere, then maybe clarifying this for the reader would be helpful. Can the authors include a similar statement (as above) in the main text clarifying that land use change causes fluxes that historically have been net positive flux and a more detailed statement in the methods?

With respect to the airborne fraction, I think that this just leads to confusion. The authors themselves state:

'we limited the discussion on the airborne fraction and its, not statistically significant, positive trend.'

If the trend is not significant, then there is no positive trend, especially if the airborne fraction trend is highly sensitive to the trends in highly uncertain underlying emissions. I would just remove the airborne fraction estimates.

Citation:

Randerson, J. T., Li, Y., Fu, W., Primeau, F., Kim, J. E., Mu, M., Hoffman, F. M., Trugman, A. T., Yang, L., Wu, C., Wang, J. A., Anderegg, W. R. L., Baccini, A., Friedl, M. A., Saatchi, S. S., Denning, A. S., & Goulden, M. L. (2025). The weak land carbon sink hypothesis. *Science Advances*, 11(37), eadr5489. <https://doi.org/10.1126/sciadv.adr5489>

Referee #5

(Remarks to the Author)

I co-reviewed this manuscript with one of the reviewers who provided the listed reports.

Response to referees' comments:

Dear Editor,

Thank you for the opportunity to revise our manuscript. The referee's comments have greatly helped to clarify our findings and we thank them for their time and insights. In particular, we have clarified and harmonised our treatment of the uncertainty across the manuscript. We note here and in the manuscript that while some of the corrections made to carbon budget components were not statistically significant, they were nevertheless meaningful because they represent known processes, and our approach has enabled the resolution of long-standing issues that had not been possible so far. The fact that the statistically significant trend, previously present in the carbon budget imbalance, is now completely resolved and indistinguishable from zero is a major advance of our manuscript. We also changed the title to a more precise one following a suggestion from referee 2. You will find below our point-by-point response.

Pierre Friedlingstein, on behalf of the author team

Referee #1 (Remarks to the Author):

Review of Friedlingstein et al 2025, submitted to Nature

Friedlingstein and colleagues present evidence to revise the approaches used in the Global Carbon Budget, which is released annually. One can expect that the adjustments to the budget proposed in this paper will be incorporated into the Global Carbon Budget going forward, making it an important contribution. However, it should be noted that the paper does not present new data, models, or analyses; instead, it summarizes many other studies and connects them within the context of the global carbon budget.

It is also important to note that the adjustments to the budget proposed are not statistically significantly different from the GCB2024 (Friedlingstein et al. 2025). Table 1 shows that while mean values have shifted, no changes are outside the previously-reported 1-sigma uncertainties. This means that there is no new understanding of the global carbon budget here. We encourage the authors to be more transparent on this issue, as it points to the lack of progress on uncertainties and the need for improved observation and modeling.

The referee is right saying that the corrections applied to the budget components are not significantly different from the GCB2024 estimates, given their uncertainty. For example, the correction on SLAND is substantial, a 0.5GtC/yr reduction, but the revised SLAND estimate 2.7 ± 0.9 GtC/yr is not significantly different from the initial estimate of 3.2 ± 0.9 GtC/yr. However, we respectfully disagree with the implication that this means there is no new understanding of the global carbon budget. The corrections we apply on the ELUC, SLAND and SOCEAN terms are all based on biogeochemical processes we do understand, but so far have not been considered in the GCB estimates. Taking the SLAND example above, it is clear that assuming pre-industrial land cover for the calculation of SLAND leads to an overestimation for the current period, as forest cover globally declined over the historical period, by about 30%. Hence our correction on SLAND of -0.5GtC, while still within the SLAND uncertainty, is an improvement in our understanding of the magnitude of the land sink. The same is true for the cool skin correction on SOCEAN estimated by the fCO₂ products. It is clear that taking sea temperature at some depth for the estimate of the CO₂ exchange between the sea surface and the air will lead to an underestimate of the flux, as the actual temperature of the skin sea surface is slightly cooler. Similarly, while the correction on SOCEAN of 0.18 GtC/yr is within the SOCEAN uncertainty (± 0.4 GtC/yr), it is an improvement in our understanding of the magnitude of the ocean sink. In the revised version of the manuscript, we clarify that while not being significantly different from the GCB2024 estimate, we are more confident in our revised estimates.

The fact that the revised estimates of the land and ocean sinks are in much better agreement with the independent estimates derived from atmospheric inversions and also from atmospheric oxygen brings additional confidence that the revised budget improves our understanding of the global carbon budget. Last, the revised budget reduces the budget imbalance, also indicating an improvement in our understanding of the global budget. We clarified this in the revised manuscript with the following new paragraphs, starting line 206: *The corrections applied to E_{LUC} , S_{LAND} and S_{OCEAN} are each within the uncertainty of the initial estimates, hence the revised estimates are not statistically significantly different from the GCB2024 estimates (Table 1). However, the corrections applied here are based on known biogeochemical processes, which have not been considered in the GCB estimates so far. Furthermore, high confidence can be placed on the sign of each of these corrections: assuming constant vegetation densities leads to an underestimation of E_{LUC} , assuming pre-industrial land cover leads to an overestimation of S_{LAND} , ignoring historical increase in lateral carbon export also leads to an overestimation of S_{LAND} , and neglecting the ocean cool skin effect leads to an underestimation of S_{OCEAN} . Hence the revised estimate of E_{LUC} , S_{LAND} and S_{OCEAN} represents an improvement in their representation in the global carbon budget. Furthermore, the revised budget, with a smaller net land CO_2 (1.4 ± 1.2 GtC/yr) and a larger ocean sink (3.1 ± 0.5 GtC/yr), is fully consistent with the estimates from atmospheric inversions (1.4 ± 0.5 GtC/yr and 3.1 ± 0.5 GtC/yr for the net land flux and the ocean sink, respectively), and with estimates derived from atmospheric O_2 observations (1.0 ± 0.8 GtC/yr and 3.4 ± 0.5 GtC/yr, respectively) (Table 1)^{13,37,49}. The convergence of these independent estimates gives stronger confidence that this revised budget provides more robust estimates compared to GCB2024.*

The budget imbalance, which was -0.4 ± 1.3 GtC/yr over 2014-2023 in GCB2024, is reduced to near zero (-0.1 ± 1.3 GtC/yr) (Fig. 1d, Table 1), although it is not statistically significantly different from the GCB2024 estimate. Finally, the statistically significant negative trend in the B_{IM} over the last 65 years of -0.14 ± 0.04 GtC/decade (p -value= 0.003) in the GCB2024 estimate is now reduced to a non-significant trend of -0.06 ± 0.04 GtC/decade (p -value= 0.14), adding confidence in the revised estimate of the global carbon budget presented here (Supplementary Fig. 3f)."

Major Comments

1. Our primary concern with the paper is the treatment of uncertainty.
 - a. At the very top level, one notes from Table 1 that the revisions to the Global Carbon Budget proposed here are not statistically significant. Mean values have changed, but none of these adjustments are outside the previously reported 1-sigma uncertainties.
 - i. The authors need to make it clear to the reader that these adjustments to the Global Carbon Budget are not statistically significant. This should be stated in the Abstract and in the Implications sections, at minimum.
 - ii. That the BIM still has large uncertainty is made clear in this version of the paper, but the authors also need to discuss the fact that these "changes in the budget" are also not significant.

i. As mentioned above, the referee is right that the corrections we apply to E_{LUC} , S_{LAND} and S_{OCEAN} are within the uncertainty of the original estimate (see Table 1). We make that clear in the revised manuscript. See new text pasted above.

ii. The uncertainty of the BIM is always calculated combining the uncertainty of all other terms of the budget quadratically. Hence it is not reduced by the corrections on the budget's components, as the uncertainty on each term is not reduced. However, the GCB2024 BIM had a trend, which indicate that there is an increasing bias in one (or more than one) of the budget components. Our revised BIM is near zero and has no significant trend over the last 65 years, which is a major improvement in our understanding and greatly reduces the likelihood of a bias. Nevertheless, as suggested by the referee, we

clarified in the revised manuscript that the revised BIM is not significantly different from GCB2024: *The budget imbalance, which was -0.4 ± 1.3 GtC/yr over 2014-2023 in GCB2024, is reduced to near zero (-0.02 ± 1.3 GtC/yr) (Fig. 1d, Table 1), although it is not statistically significantly different from the GCB2024 estimate.*

b. All of the bar plots in the paper should include uncertainty bars. At present, these are found only on Figure 3 and 4. They are needed on Figure 1 and S6.

Done, thank you

i. Where figures are labeled with quantitative values, uncertainties should be included. These figures have lots of white space of which to make use.

Done, thank you

c. All numbers in the text should include uncertainty. Only some do in this version.

Thank you, we quantified the uncertainty on all terms of the revised budget.

For the ELUC and SLAND corrections, the uncertainties are estimated across the range of studies/models available. For the SOCEAN correction, our quantified uncertainty is based on available studies augmented by our expert judgement. We assess our confidence in each correction to be positive. We assess that it is “very likely”, i.e. 90% chance that the GOBM correction is positive, hence we can derive a standard deviation consistent with a 10% chance the correction is negative (Z-score = -1.28). For the fCO₂-product correction, we assess it is “likely”, i.e. 68% chance that the GOBM correction is positive, hence we can derive a standard deviation consistent with a 32% chance the correction is negative (Z-score = -0.45).

The uncertainty estimate on each term of the revised budget is described in more details in the Methods section.

2. A second concern is the lack of clarity in the Methods regarding the actual adjustments made. The authors state justifications for adjustments that imply the approach taken, but do not consistently make clear statements about precisely what was done, e.g. “GOBM air-sea fluxes as reported were increased by 10%”. We suggest a short summary at the end of each Supplementary section to explicitly state the adjustments that are made for each component of the budget.

As suggested by the referee, we harmonised the method section, with a short summary at the end of each section which explicitly states the adjustment done (GtC/yr) on the component of the budget. We also expanded the discussion on the quantification of uncertainty.

Minor comments

- We note that the authors neglect to include page and line numbers in the manuscript, adding a burden for Reviewers. We use the headers of the sections here. The authors are requested to please include page and line numbers in future versions of their manuscript.

Sorry for that omission, done in the revised manuscript.

- When describing results, please refer to specific figure panels (e.g., Fig. 2a) when appropriate.

Done, thank you

Abstract

- “continue” not “continued”

Done, thank you

- “sinks, contributing”

Done, thank you

- Please include an explicit statement that the revisions to the mean values of the budget terms

are not statistically significantly different from previous estimates given their 1-sigma uncertainties.

Introducing latest evidence

- Please explain why the authors do not revise to the pre-industrial land-to-ocean export of 0.55 ± 0.1 PgC/yr published by Liu et al. 2024 in Nature Geoscience.

Note that the specific value of the preindustrial export is not the focus of our study here which aims to quantify the historical perturbation on the lateral carbon export. Also, we note that the Liu et al estimate of 0.55 ± 0.1 for the preindustrial export is not statistically different from the GCB2024 estimate of 0.65 ± 0.3 GtC.

However, we do cite Liu et al in the Methods section describing the LCE correction as their present day estimate of the carbon export is close to our estimate.

- Cite uncertainty on 0.34 GtC/yr

Uncertainty on this number has been added in the Methods section: *“The anthropogenic perturbation (2014-2023 minus pre-industrial) on the lateral land-to-inland water carbon flux (F'_{LI}) amounts to 0.54 ± 0.44 GtC/yr and is partitioned into increased aquatic CO_2 evasion (F'_{IA} , 0.34 ± 0.26 GtC/yr), aquatic carbon storage (F'_{IS} , 0.09 ± 0.03 GtC/yr), and carbon exports to the ocean (F'_{IE} , 0.11 ± 0.08 GtC/yr).”*

- It is not clear what is meant by “uniquely”

Removed in the revised manuscript

- Cite uncertainty on 3.2 GtC/yr 2.6 GtC/yr

Done, thank you, new sentence is:

However, fCO_2 -products suggest a substantially larger ocean sink than GOBMs (3.1 ± 0.3 GtC/yr versus 2.6 ± 0.4 GtC/yr, respectively, over 2014-2023)

- Add reference after “atmospheric carbon and oxygen” to Keeling paper

Done, thank you.

- Please explain ‘statistical differences’ - the term and the use of quotations

Term removed

- Strike “allow to determine” and replace with “supporting determination of”

Sentence rephrased

Revised estimates of the global carbon budget

- “(p-value = 0.003; Supplementary Fig. 3)”

We rewrote the sentence. To be consistent with GCB2024, we estimate the trends in ELUC since the 1990s: *Although the correction increases land-use change emissions with time, the statistically significant decline in E_{LUC} of 0.2GtC/decade since the late 1990s, as identified in GCB2024, remains (p-value<0.001).*

Land use change correction

- CO_2 , not CO_2

Done, thank you

Land sink replaced sinks and sources correction (RRS)

- Consider re-writing the header

Done, thank you

- Consider re-writing this sentence: “This issue has been referred to as the loss of sink capacity is called here the replaced sinks and sources (RRS)”

Done, thank you, sentence now reads as: *This issue is known as the replaced sinks and sources (RSS)^{21,24} (in some publications also called the loss of sink capacity²³).*

Land sink lateral carbon export correction

- For consistency, use either “machine-learning” or “machine learning”

Done, thank you, we use “machine learning”

Atmospheric CO₂ growth rate from surface concentrations

- “This results in inverse-based annual estimates of the CF (Supplementary Fig. 5c)”: Should this be 5a and not 5c?

Done, thank you

Airborne fraction

- This is the first time Supplementary Figure 2 is mentioned. Perhaps move/re-number this figure

Done, thank you. That figure is now referred to in the introduction, along with the discussion on the trend in the BIM.

- BIM, not BIM

Done, thank you

Climate change impact on the global carbon budget

- CO₂, not CO₂

Done, thank you

Methods

Land use Change correction

- Define TRENDYS2 / TRENDY S2

Sentences rephrased, the TRENDY acronym has been removed.

- State explicitly the adjustments made

Land Sink replaced...

Done, for all corrections we added a summary sentence which states explicitly the adjustment made on the budget component. For SLAND RSS it reads:

This correction leads to a decrease of S_{LAND} by 0.5 ± 0.3 GtC/yr for the 2014-2023 period.

- Strike “as we have shown”

Done, thank you

- State explicitly the adjustments made

Land sink lateral ..

Done. Summary sentence added : *“In summary, the lateral carbon export (LCE) correction leads to a 0.07 ± 0.06 GtC/yr reduction of S_{LAND} , with the uncertainty estimated by combining the uncertainties reported in the original studies for enhanced CO₂ outgassing^{26,27}. No LCE correction on S_{OCEAN} was applied here.”*

- Replace “largely relying on observations and machine learning” with “relying on process models, observations and machine learning”

Done, thank you

- Include uncertainty on all numbers

Done, thank you, uUncertainty added on all numbers.

- State the time frame of analysis

Thank you, the time frame of analysis, 2014-2023, has been added in the methods and also in Figure S4

- Explain better what justifies the choice to not include the 0.11 GtC adjustment in the ocean, i.e. justify the choice to assume this carbon is stored

Done, sentence added: *“We do not correct the GCB estimate of the ocean sink (S_{OCEAN}), i.e., we assume that the terrestrial carbon exported to the ocean (F'_{IE} , 0.11 ± 0.08 GtC/yr GtC/yr) remains stored in the ocean, as the fate of the land-derived carbon in the coastal and open ocean remains too uncertain to be quantified with confidence ²⁶.*

Such an assessment could only be achieved through modelling and no ocean biogeochemical model has yet assessed how the land-to-ocean perturbation on the carbon cycle impacts the open ocean air-seas CO₂ exchange (only the impact of land-derived anthropogenic nutrient inputs has been assessed).

- State explicitly the adjustments made

Done, summary sentence added: *In summary, the lateral carbon export (LCE) correction leads to a 0.07 ± 0.06 GtC/yr reduction of S_{LAND} , with the uncertainty estimated by combining the uncertainties reported in the original studies for enhanced CO₂ outgassing ^{26,27}. No LCE correction on S_{OCEAN} was applied here.*

Ocean Sink

- Include uncertainty on all numbers

Done, thank you.

- State explicitly the adjustments made

Done, summary sentence added: *“In our revised assessment, we increase the GOBMs estimate by $10 \pm 8\%$ and the fCO_2 -products estimate by 0.18 ± 0.4 GtC/yr. These two corrections combined lead to an increase of S_{OCEAN} by 0.22 ± 0.23 GtC/yr for the 2014-2023 period.”*

- Add a reference for the sentence beginning “Correcting the GOBM...”

This sentence has now been removed as unnecessary.

Atmospheric CO₂...

- Explain in a few sentences the method of van der Woude et al. Readers should be able to understand this to first-order without having to read another paper.

Done, we explained it in the next two sentences, rewritten for clarity: *“We use the model-sampled mole fractions at the surface stations to calculate the annual CO₂ growth rate (in ppm/yr), following the same calculation for the observations as developed by ⁴⁸, similar to the method used by the National Oceanic and Atmospheric Administration (NOAA) ⁴⁶. We calculate the annual net input of CO₂ in the atmosphere (in GtC/yr) as the sum of the annual fossil fuel emissions and the inverse-derived net land and ocean sinks. The annual ratio of this net annual input of CO₂ divided by the annual growth rate gives the CF (in GtC/ppm). This is repeated for each inverse model and results in annual estimates of the CF (Supplementary Fig. 5a), with their standard deviation. ”*

- State explicitly that no adjustments have been made

Done, summary sentence added : *“No adjustment on G_{ATM} itself is made here as the year to year changes in CF need further evaluation.”*

Figure 1 is difficult to understand, and it is difficult to see the adjustments being proposed.

- Uncertainty bounds needed

Done, thank you

- Colors should be consistent with Figure 3 for each term

Done, thank you

- The Atmosphere and FF terms should be indicated somehow; explain abbreviations in the figure caption (dL, RSS, LCE)

GATM and EFOS are unchanged here and hence only shown in Figure 3 (revised budget), but not in figure 1 (consolidated components). dL, RSS and LCE are now explained in the caption, thank you

- Please include subpanel labeling and refer to these in text

Done, thank you

- Please increase font size

Done, thank you

- There is a lot of white space here that can be used to improve this figure; one of the most interesting aspects about this figure is the difference between GCB2024 and this study, but these differences are very difficult to see

Done, thank you

- Add a, b, c, d

Done, thank you

- Add GtC/yr label

Done, thank you

- Consider combining this figure with Figure 3. This could show GCB2024 in top row, adjustments (with reduced vertical extent) in middle, and final in bottom. All with uncertainty bounds. This will help the reader more clearly understand the adjustments being made.

We don't think that merging Figure 1 and Figure 3 would work, as figure 1 shows the corrections to the ELUC, SLAND and SOCEAN components, and impact on the BIM, while figure 3 shows the global budgets (all components), including the separate effects of CO2 and climate change on the ocean and land sinks. A combined figure with 3 panels would probably be too confusing for the reader.

Table 1/S1

- There is a lot of white space here that can be used to improve these tables. It would also be useful to add a row that shows the difference between GCB2024 and this study. We suggest something like: (see attached Picture1.png)

We improved Table 1 as suggested. Thank you.

Figure 3

- Use the appropriate subscript for the x-axis labels (EFOS, ELUC etc.)

Done, thank you

- Indicate which components have been updated in this study; e.g., bold x-axis label, or use thicker box edge lines

Done, thank you, updated components are indicated with dashed outlines.

- Change figure description to: "Revised Global Carbon Budget"

Figure title changed to "Consolidated global carbon budget"

- It would be useful to compare to GCB2024; could you add boxes for the GCB2024 in the

background?

- As noted above, consider combining with Figure 1 to make a single figure

As explained above, we don't think that would work, figure 1 and figure 3 focusing on different elements (changes in individual components vs new consolidated budget and sinks attribution to drivers).

Figure S2

- Remove white space

Done, thank you

Figure S4

- Caption needs to indicate what gray box is, and what the dates are

Done , gray box is Fossil (EFOS), dates are 2014-2023. All added to the figure.

- The boxes need to be labeled (ATM, Ocean, etc)

Done

- (b) has 0.09 and 0.11 switched from where they should be based on the text

The text in methods has been corrected, thank you

- Increase font size

Done

- Consistently use the same labels (Fie, FLI, etc.) as in the caption and text on the figure.

Done

- Include uncertainties on all numbers

Done, thank you for all suggestions

Figure S6

- Uncertainty bounds are needed

Figure S6 has been removed as unnecessary.

Figure S7

- Remove white space

Figure S7 has been removed as we removed most of the discussion on the airborne fraction, as suggested by referee 2.

Figure S8

- CO₂, not CO₂

Done, thank you

Figure S9

- Please label the lines with a legend

Figure S9 has been removed as we removed most of the discussion on the airborne fraction, as suggested by referee 2.

- The text suggests that the blue includes the impact of climate change, but the caption says this study is black. Both numbers appear to derive from this study. Please clarify.

Figure S9 has been removed.

[please also see file attached]

Referee #2 (Remarks to the Author):

I co-reviewed this manuscript with one of the reviewers who provided the listed reports.

Referee #4 (Remarks to the Author):

Title: Trends in sources and sinks of carbon dioxide over the past 65 years.

Summary:

- 1) Here the authors make minor adjustments to the Global Carbon Budget from 2024 (GCB2024), by altering land use emissions and how they are treated in models, that appear to have a big impact in reducing the net land C sink. The question is whether this study really advances our understanding of the global carbon budget (beyond that of GCB2024) or just found a way to explain the residual variance not predicted by land models. We are not convinced given the uncertainty of land use change fluxes and their potential impact on questionable trends in the airborne fraction. Furthermore, given these uncertainties in the global fluxes, updates to regional fluxes may be less credible, or at least need further discussion.

In our study, we corrected not just ELUC but also SLAND and SOCEAN, and we considered carbon fluxes from anthropogenic influences on lateral carbon export. All these corrections are based on improved process understanding of the global carbon cycle that has not yet been reflected in earlier GCB estimates. Regarding land fluxes, the inclusion of δL in the ELUC estimate and of RSS in the SLAND estimate make the ELUC and SLAND estimates more consistent as both of them now consider the impacts of environmental effects and of anthropogenic land cover changes. Note that this does not change the definitions of ELUC and SLAND, with the former still exclusively quantifying fluxes from land-use change and the latter the natural land sink. For the most recent decade, the corrections increase ELUC by +0.1 GtC/yr and decrease SLAND by -0.5 GtC/yr (see Table 1). As the net land sink is SLAND-ELUC, the biggest impact in reducing the net land C sink comes from the change in the land sink rather than from “altering the land use emissions”, which only accounts for $0.1/0.6=16\%$ of the reduction in the net land C sink. For SOCEAN, the improvements are two-fold. First we incorporate a known process that was missing in data-product estimates based on ocean fCO_2 observations. Second we reassess the ocean sink from the process models based on independent evaluation of model results using a range of observations.

It is correct that CO_2 emissions from land-use change are uncertain, as ELUC is the most uncertain flux of the global carbon budget (in relative terms). It is also correct that the uncertainty of the budget (and in particular of the BIM) limits our ability to attribute changes in the airborne fraction to changes in individual GCB components as we show in Supplementary Figure 2. Despite the persisting substantial uncertainties in ELUC and SLAND, our improvements based on process understanding represent an important step forward in the quantification of the global carbon budget.

In light of the comments of the referee, we limited the discussion on the airborne fraction and its, not statistically significant, positive trend. While the reduced trend in the BIM brings confidence that the corrections on the budget components applied here are important, the irreducible uncertainty of ELUC highlights the importance of being cautious when analysing changes in the airborne fraction over time. Hence, we now limit the discussion on the airborne fraction to Supplementary Figure 2, to illustrate that the trend in the BIM precludes any robust analysis of the long-term trends in the components of the global carbon budget. We thank the referee for raising that point here.

- 2) General Comments:

In general, land use change (LUC) fluxes have the greatest relative uncertainty in the carbon

budget (~50%) and therefore any changes in land use and how they are treated can have large impacts on inferred trends in the global land C sink. Land use changes can lead to changes in CO₂ two-way fluxes-not just one-way emissions. The authors here assume that all land use changes lead to a positive emission of CO₂ to the atmosphere; however, there are instances where LUC over time can lead to negative CO₂ fluxes from the atmosphere. For instance, historic land use change over the central North America undoubtedly lead to a release of CO₂ to the atmosphere when primary forests were converted to grazing and farming lands. However, the intensive agricultural land use occurring today over central North America explains most of the continental net land CO₂ sink (Liu et al., 2018). Thus land use over parts of North America appears to have transformed ecosystems from CO₂ sources and then back to CO₂ sinks over time.

The bookkeeping models used here, and in the GCB, to estimate ELUC do account for the fact that anthropogenic land use can lead to both emissions (e.g. from deforestation, decay of wood products and decay of slash after wood harvest) and removals (e.g. from afforestation and reforestation, regrowth after wood harvest), as shown in Figure 7 of GCB2024 (reproduced here as Figure R1, see panel (c)). As the referee mentions correctly, parts of the world including North America, Europe, and China are currently net carbon sinks from land-use change (see Figure 6b of GCB2024, <https://doi.org/10.5194/essd-17-965-2025>), mainly due to forest regrowth. However, for decades and when assessed on the global scale, the net effect of anthropogenic land-use change has been a source to the atmosphere. The reported global net ELUC estimates are thus emissions (see panel (a) of Figure R1).

Regarding North America, it is correct that land-use change caused emissions due to deforestation in the USA (up to ~1960s), whereas in the last 50-60 years net ELUC in the USA was close to 0 or even a sink according to the bookkeeping model estimates.

Figure R1: ELUC with sub-components (reproduced from GCB2024, Figure 7 there).

- 3) Because there is so much uncertainty associated with mean annual land use fluxes any trends in land use fluxes over time should be interpreted with caution. For instance, these very same co-authors have previously reported apparent increasing trends in land use fluxes (Hong et al., 2021), in contrast to the decreasing trends reported here (Fig. S3, regardless of C density).

Indeed, much care is needed when reporting on trends in land-use fluxes given their large uncertainty. In the GCB papers we have always been extremely cautious on interpreting the recent trends in ELUC. Until GCB2023, we accompanied the ELUC estimates in the GCBs by the following statement: “A small decrease over the past 2 decades is not robust given the large model uncertainty”. In GCB2024, a revised land cover forcing over China led to a strong increase in land-use removals in China (as in the meantime also evidenced by the study from Zhu et al., 2025, <https://doi.org/10.1038/s41558-025-02296-z>), leading to a statistically significant decreasing trend in global ELUC over the last one, two decades. Revised and refined land cover data was also introduced for Brazil (in 2022) and Indonesia (in 2023), currently the two biggest emitters in terms of ELUC. Hence, in GCB2024, we were confident enough to report for the first time “Since the late 1990s, emissions from LULUCF have shown a statistically significant decrease at a rate of around 0.2 GtC per decade.”

We confirm here that the decline in net ELUC remains statistically significant also after accounting for transient carbon densities since the late 1990s. We have rephrased the corresponding part in the manuscript accordingly: *The revised estimate of ELUC, when*

accounting for transient carbon densities, is 1.2 ± 0.7 GtC/yr for the last decade (2014-2023), which is 0.11 ± 0.04 GtC/yr higher than in GCB2024. Although the correction increases land-use change emissions, the statistically significant decline in ELUC since the late 1990s, as identified in GCB2024, remains (p -value <0.001).

The positive trend in the study by Hong et al. (2021) is attributable to the combination of including non-CO₂ greenhouse gases and relying on earlier estimates of land-use change CO₂ emissions (GCB2019), which had not yet incorporated improvements in the land-use change forcing in China and other corrections and updates in land-use forcing, in particular in Brazil and Indonesia. Also, Hong et al. used only one bookkeeping model, a comparison to the other bookkeeping models (Extended Figures 5-7 in Hong et al.) had highlighted the uncertainties around the trend estimates.

- 4) Furthermore, previous analyses have investigated how slight changes in land use emissions associated with deforestation in particular can affect our estimates of the land and ocean C sinks over time as indicated by the airborne fraction (van Marle et al., 2022) and this paper was retracted due to dubious statistics. Based on this previous retraction, it is surprising that the authors appear to rely on simple correlations of the airborne fraction over time, rather than more robust Mann-Kendall trend tests or Monte-Carlo simulations (Bennedsen et al., 2023). Lastly, the authors selectively cite previous studies reporting significant trends in the airborne fraction that are somewhat dated and dubious at best. More recent detailed analyses of the airborne fraction show no significant trend and considerable variability in apparent trends depending upon the land use change emissions considered (Bennett et al., 2024).

As mentioned above, we agree with the referee that analysis of the trend in the airborne fraction is complex and should include a detailed analysis of the uncertainty on both the fossil fuel and the net land use CO₂ emissions estimates. Also, the year-to-year variability of the airborne fraction is large, primarily due to the large variability of the land sink. Hence, the trend detected in airborne fraction is not significant, as was already acknowledged in the submitted manuscript. Hence, we removed the discussion on the airborne fraction trend and the specific role of climate change, keeping the focus of the paper on the consolidated budget and the effect of climate on the carbon sinks and atmospheric CO₂.

- 5) The way in which the current results are presented makes it unclear whether the main conclusion is that the land carbon sink has always been overestimated, or that the land sink is starting to saturate and taking up less carbon relative to the ocean. Although there are subtle differences in these conclusions, the implications of these different conclusions are quite substantial and not really distinguished in the conclusions. For instance, the conclusion in the abstract that- ‘The combined effects of climate change and deforestation turns South American and Southeast Asian ecosystems from CO₂ sinks to sources.’ - seems to be at direct odds with recently updated forest inventory data that show a persistent land sink due in part to increases in forest plot density and tropical forest regrowth (Pan et al., 2024). Is secondary forest regrowth considered at all in these revised land use change flux estimates?

Thank you, we tried to make it clearer in the revised manuscript. The net land sink has always been overestimated in previous GCB due to the replaced sources and sinks (RSS) issue. That is something we acknowledged in all past global carbon budget ESSD papers (it was referred to as the loss of additional sink capacity, of which RSS is the dominant part). Our revised budget presented here corrects for this issue. Together with the correction on SOCEAN, we now find that the land sink is lower than the ocean sink (2.7 ± 0.9 vs 3.1 ± 0.5 GtC/yr). This has nothing to do with any saturation of the sinks.

Our findings are consistent with Pan et al. 2024 who reported: “Although the carbon sinks in tropical intact and regrowth forests were large, high emissions resulting from deforestation and degradation counteracted nearly all of these remarkable sinks, making tropical forest lands almost carbon neutral (Extended Data Fig.3), with a small net sink or source of between -0.1 and 0.6 Pg C yr⁻¹, fluctuating with deforestation intensities in different decades.”.

The Extended Data Fig. 3 of Pan et al. 2024 show that tropical forests are a small sink of 0.29 ± 0.63 GtC/yr when accounting for deforestation, consistent with our findings. We also note that the Pan et al. estimate of the net land sink (1.6 GtC/yr for the 2010s, presented in their Figure 2) is consistent with our estimate of 1.4 ± 1.1 GtC/yr for the last decade (see Table 1).

- 6) Lastly, if the argument is that the land portion of the global carbon sink has diminished over time, then there are several recent satellite records that may corroborate this finding (Li et al., 2023; Wang et al., 2020) and could be used to put these results into context, especially at the regional scale. However, these studies were not cited.

Thank you. In addition to Brienen, et al. 2015, Hubau, et al. 2020, and Gatti, et al. 2021, we have added a reference to Li et al. 2023, new ref 57, when reporting our results on the negative effect of climate change on ecosystem CO₂ uptake in the tropics. We do not cite Wang et al. 2020, as they discuss the potential saturation of the CO₂ fertilisation effect, something we did not investigate here.

- 7) The regional characteristics of LUC flux and its contribution to the updated global carbon budget need better explanation. Figure 3 displayed the LUC flux at a global scale, and Figure 4 displayed regional contributions. However, It would be good to see how updated LUC fluxes affect the revised global and regional carbon budgets, and which region(s) dominate the budget difference between GCB2024 and the revised GCB. It is possible a few particular regions dominate the revised GCB and budget balance.

Thank you for this suggestion. We have estimated the regional contributions and added a sentence to the manuscript: “About 75% of the increase in ELUC is due to larger net land-use change emissions in the regions South America, Southeast Asia, and Africa.”

- 8) The increase in LUC flux in this revised GCB estimate appears to be largely due to the use of transient carbon density, rather than the land use data itself. In this case, the transient model estimated a higher transient carbon density than static carbon density in GCB 2024, such that there is higher carbon emission per unit area in land use change. Correct? If so, why is there a higher transient carbon density? Is this density of the biomass (gC/m³) or is it ecosystem density (gC/m²) as estimated by Pan et al. 2024 that is necessary to solve for the persistent forest C sink? Is this due to changing environmental factors, such as atmospheric CO₂ and climate, or does the dynamic global vegetation modeling capture higher growth? Furthermore, is this transient carbon density incorporated into the ensemble of models considering the replaced sinks and sources (RSS) and if so doesn't this suggest that 'land use emissions' are just a subset of the net land flux term?

The referee is correct that the revised upward estimate of ELUC is entirely due to the accounting for transient ecosystem carbon densities (i.e. biomass carbon plus soil carbon densities, all in gC/m²). Carbon densities increase over the historical period due to environmental effects, primarily the CO₂ fertilisation effect due to the atmospheric CO₂ increase in particular in forest ecosystems. Effects from climate change (which may be detrimental to land carbon stocks) and nitrogen deposition are also included in our estimates of changes in carbon densities, as the transient carbon densities are estimated from DGVM simulations which account for all of these environmental factors. The combined effect of these environmental changes leads to larger net ELUC emissions (due to larger carbon densities) in the last few decades. Details, including robustness of these statements across models, can be found in Dorgeist et al. (2024), supplementary figures 3 and 4. We note that Pan et al. 2024 also uses transient carbon densities that are increasing over time as in our study (see their Extended Data Fig. 2c).

The RSS correction for SLAND addresses a separate issue. It accounts for lost forest area since 1700 (about 20% reduction), and the resulting loss of the natural forest carbon sink, when estimating SLAND with DGVMs.

The net land flux term is obtained as the difference between SLAND (including the RSS correction) estimated by DGVMs, and ELUC (including the dL correction) estimated by bookkeeping models. The net land flux is also the flux that is inferred by atmospheric inversions and atmospheric O₂ derived budgets (see Table 1).

9) Specific Comments (see comments in PDF):

The title is not very compelling and belies the abstract- maybe something revealing the discovery here.

Thank you, we propose a new title that reveals the main findings: “Emerging climate impact on carbon sinks in a consolidated carbon budget”.

10) Be consistent with units. Some figures use PgC (preferred) and others GtC (less preferred).

Done, thank you. We use GtC, preferred to be consistent with GCB2024.

References:

- Bennedsen, M., Hillebrand, E., & Koopman, S. J. (2023). On the evidence of a trend in the CO₂ airborne fraction. *Nature*, 616(7956), E1–E3. <https://doi.org/10.1038/s41586-023-05871-6>
- Bennett, B. F., Salawitch, R. J., McBride, L. A., Hope, A. P., & Tribett, W. R. (2024). Quantification of the airborne fraction of atmospheric CO₂ reveals stability in global carbon sinks over the past six decades. *Journal of Geophysical Research. Biogeosciences*, 129(3). <https://doi.org/10.1029/2023jg007760>
- Hong, C., Burney, J. A., Pongratz, J., Nabel, J. E. M. S., Mueller, N. D., Jackson, R. B., & Davis, S. J. (2021). Global and regional drivers of land-use emissions in 1961–2017. *Nature*, 589(7843), 554–561. <https://doi.org/10.1038/s41586-020-03138-y>
- Li, F., Xiao, J., Chen, J., Ballantyne, A., Jin, K., Li, B., Abraha, M., & John, R. (2023). Global water use efficiency saturation due to increased vapor pressure deficit. *Science*, 381(6658), 672–677. <https://doi.org/10.1126/science.adf5041>
- Liu, Z., Ballantyne, A. P., Poulter, B., Anderegg, W. R. L., Li, W., Bastos, A., & Ciais, P. (2018). Precipitation thresholds regulate net carbon exchange at the continental scale. *Nature Communications*, 9(1), 3596. <https://doi.org/10.1038/s41467-018-05948-1>
- Pan, Y., Birdsey, R. A., Phillips, O. L., Houghton, R. A., Fang, J., Kauppi, P. E., Keith, H., Kurz, W. A., Ito, A., Lewis, S. L., Nabuurs, G.-J., Shvidenko, A., Hashimoto, S., Lerink, B., Schepaschenko, D., Castanho, A., & Murdiyarso, D. (2024). The enduring world forest carbon sink. *Nature*, 631(8021), 563–569. <https://doi.org/10.1038/s41586-024-07602-x>
- van Marle, M. J. E., van Wees, D., Houghton, R. A., Field, R. D., Verbesselt, J., & van der Werf, G. R. (2022). New land-use-change emissions indicate a declining CO₂ airborne fraction. *Nature*, 603(7901), 450–454. <https://doi.org/10.1038/s41586-021-04376-4>
- Wang, S., Zhang, Y., Ju, W., Chen, J. M., Ciais, P., Cescatti, A., Sardans, J., Janssens, I. A., Wu, M., Berry, J. A., Campbell, E., Fernández-Martínez, M., Alkama, R., Sitch, S., Friedlingstein, P., Smith, W. K., Yuan, W., He, W., Lombardozzi, D., ... Peñuelas, J. (2020). Recent global decline of CO₂ fertilization effects on vegetation photosynthesis. *Science*, 370(6522), 1295–1300. <https://doi.org/10.1126/science.abb7772>

[please also see file attached]

Referee #5 (Remarks to the Author):

I co-reviewed this manuscript with one of the reviewers who provided the listed reports.

Here are our responses to the comments (from referee 1 or 4) made in the pdf of the manuscript

- **Abstract: the first sentence is primarily for context.**
- **Abstract: although we sympathise with referee 1, we would rather keep “net emissions from land use” to emphasize that this is the net balance between land use emissions and land use removals, as referee 2 seemed to assume we only account for the emissions.**
- **Abstract: indeed, the effect of climate change on the sink is relative to the potential, CO₂-only sink. This is explained in the Methods section. Probably not appropriate for the abstract.**
- **Intro, comment on mass conservation. There is no reason why mass conservation should be perfect in the GCB as all components are estimated independently. Also, there is also no reason to assume the land models are the sole cause of the budget imbalance. Here we show that the GCB2024 estimate of the ocean sink was biased low, also contributing to the budget imbalance.**
- **Revised estimate. We now state in the abstract that the ocean sink is 15% larger than the land sink. Thank you.**
- **Influence of climate change. The impact of climate change is diagnosed from the GCB2024 DGVMs and GOBMs and is added to the revised estimates of SLAND and SOCEAN from this study. Doing so, we assume that the corrections on the budget components are not substantially affecting the isolated effects of climate change (such as changes in temperature, precipitation, ocean circulation) on the sinks.**
- **All other minor comments have been addressed already.**

Response to Reviewers comments

Referee #1 (Remarks to the Author):

Based on a variety of recent evidence, Friedlingstein and colleagues suggest revisions to the datasets supporting the annual Global Carbon Budget. With these adjustments applied, the mean of the Budget Imbalance is reduced to near zero. As these adjustments to the budget can be expected to be directly incorporated into the Global Carbon Budget going forward, it is an important contribution.

We have reviewed the revised manuscript and find that the authors have adequately addressed all our comments from the first round of review. We recommend publication.

Thank you

Referee #4 (Remarks to the Author):

Friedlingstein et al. Second Review

We appreciate the authors revisions to the previous draft of this paper and the depth of their knowledge on this topic. However, we still have concerns whether their revised global carbon budget advances our insight into global carbon cycle processes, or simply reconciles model simulations with observations.

Much of their analysis has focused on reconciling the apparent imbalance between the inferred land C sink and the model simulated land C sink, which is a persistent problem that has vexed C cycle scientists for some time. In fact, in their response they state-

‘The fact that the statistically significant trend, previously present in the carbon budget imbalance, is now completely resolved and indistinguishable from zero is a major advance of our manuscript.’

I agree that this is a major contribution; however, I would also encourage them to be a bit more cautious. They are concluding that they have been able to perfectly match theory (ie. models) with observations (ie. atmospheric/ocean inversions). If this is true then there is no need for further terrestrial model development, but this is not true because we know that terrestrial models are still missing critical processes, such as non-respiratory C loss pathways and lateral C transport. Furthermore recent analyses have even questioned whether the inferred C land sink is overestimated and propose other potential solutions to this problem (Randerson et al., 2025). Perhaps a statement at the conclusion of the paper noting that they have been able to reconcile the ‘net land C sink’ but that future model development is necessary to advance our understanding of the underlying processes.

The paragraph about remaining uncertainties (section Implications) has been expanded, following the reviewer suggestion :

“Important uncertainties remain, as reflected by the large interannual variability still present in the BIM, and global agreement between bottom-up and top-down estimates could still be due to compensating errors in critical processes in components of the global carbon budget.”

The authors acknowledge that land use change fluxes can be positive or negative. In fact, one of the authors has a paper on the uncertainty of land-use fluxes (Pongratz et al. 2014) and thus referring to them strictly as emissions is misleading:

‘parts of the world including North America, Europe, and China are currently net carbon sinks from land-use change (see Figure 6b of GCB2024, <https://doi.org/10.5194/essd-17-965-2025>), mainly due to forest regrowth. However, for decades and when assessed on the global scale, the net effect of anthropogenic land-use change has been a source to the Atmosphere.’

The authors are following the standard in the literature, but when the standard is misleading and cause estimates to be biased towards positive fluxes to the atmosphere, then maybe clarifying this for the reader would be helpful. Can the authors include a similar statement (as above) in the main text clarifying that land use change causes fluxes that historically have been net positive flux and a more detailed statement in the methods?

Thank you, sentence added:

Note that while the net effect of anthropogenic land-use change is a source of CO₂ to the atmosphere, parts of the world including North America, Europe, and China are currently net carbon sinks from land-use change.

With respect to the airborne fraction, I think that this just leads to confusion. The authors themselves state:

‘we limited the discussion on the airborne fraction and its, not statistically significant, positive trend.’

If the trend is not significant, then there is no positive trend, especially if the airborne fraction trend is highly sensitive to the trends in highly uncertain underlying emissions. I would just remove the airborne fraction estimates.

Thank you, the sentence on the airborne fraction and supplementary Figure 2 have been removed.